# Urban population exposure to NO$_X$ emissions from local shipping in three Baltic Sea harbour cities – a generic approach

Martin Otto Paul Ramacher[1], Matthias Karl[1], Johannes Bieser[1], Jukka-Pekka Jalkanen[2], Lasse Johansson[2]

[1]Chemistry Transport Modelling Department, Institute of Coastal Research, Helmholtz-Zentrum Geesthacht, 21502, Geesthacht, Germany
[2]Finnish Meteorological Institute, P.O. Box 503, 00101 Helsinki, Finland

*Correspondence to*: Martin Ramacher (martin.ramacher@hzg.de)

**Abstract.** Ship emissions in ports can have a significant impact on local air quality (AQ), population exposure, and therefore human health in harbour cities. We determined the impact of shipping emissions in harbours on local AQ and population exposure in the Baltic Sea harbour cities Rostock (Germany), Riga (Latvia) and the urban agglomeration of Gdansk-Gdynia (Poland) for 2012. An urban AQ study was performed using a global-to-local Chemistry Transport Model chain with the EPISODE-CityChem model for the urban scale. We simulated NO$_2$, O$_3$ and PM concentrations in 2012 with the aim to determine the impact of local shipping activities to population exposure in Baltic Sea harbour cities. Based on simulated concentrations, dynamic population exposure to outdoor NO$_2$ concentrations for all urban domains was calculated. We developed and used a novel generic approach to model dynamic population activity in different microenvironments based on publicly available data. The results of the new approach are hourly microenvironment-specific population grids with a spatial resolution of $100 \times 100$ m². We multiplied these grids with surface pollutant concentration fields of the same resolution to calculate total population exposure. We found that the local shipping impact on NO$_2$ concentrations is significant, contributing with 22%, 11%, and 16% to the total annually averaged grid mean concentration for Rostock, Riga and Gdansk-Gdynia, respectively. For PM$_{2.5}$, the contribution of shipping is substantially lower with 1-3%. When it comes to microenvironment-specific exposure to annual NO$_2$, the highest exposure to NO$_2$ from all emission sources was found in the home environment (54-59%). Emissions from shipping have a high impact on NO$_2$ exposure in the port area (50-80%) while the influence in home, work and other environments is lower on average (3-14%), but still with high impacts close to the port areas and downwind of them. Besides this, the newly developed generic approach allows for dynamic population weighted outdoor exposure calculations in European cities without the necessity of individually measured data or large-scale surveys on population data.

## 1 Introduction

According to the International Maritime Organization (IMO), more than 90% of world trade is carried by sea since maritime transport is the most cost-effective way to move mass goods and raw materials (International Maritime Organization,

2015). However, maritime transport is an important source of air pollutants on the global (Wang et al., 2008) and European level (Eyring et al., 2010) and can contribute significantly to local air quality (AQ) problems in European harbour cities of all sizes (Viana et al., 2009). Globally, ships are known to emit 5–7 $10^9$ kg yr$^{-1}$ of nitrogen oxides (NO$_x$), 4.7–6.5 $10^9$ kg yr$^{-1}$ of sulphur dioxide (SO$_2$), and 1.2–1.6 $10^9$ kg yr$^{-1}$ of particulate matter (PM) into the atmosphere (Smith et al., 2014; Corbett and

Koehler, 2003; Eyring et al., 2005). Seventy percent of these emissions occur near coastlines and therefore contribute to air pollution in both coastal areas and harbour cities (Andersson et al., 2009; Corbett et al., 1999; Endresen, 2003). Ships emit NO$_x$ mainly in the form of NO, which is quickly converted to NO$_2$, thus atmospheric NO$_x$ from shipping is mainly in the form of NO$_2$ (Eyring et al., 2010). The contribution of international shipping to the air quality over European Seas reached up to 80% for NO$_x$ and SO$_2$ concentrations up to 25% for particles with a diameter of 2.5 µm and less (PM$_{2.5}$) and up to 15% for

ozone (O$_3$) in hotspot areas along coastlines in 2005 (EEA, 2013). In the North Sea region, the relative contribution of international shipping to NO$_2$ concentration levels ashore close to the sea can reach up to 25% in summer and 15% in winter (Aulinger et al., 2016), while Karl et al. (2018) showed average shipping contributions of 40% over the Baltic Sea and 22–28% for the entire Baltic Sea region. In the entire Baltic Sea region the average contribution of ships to PM$_{2.5}$ levels is in the range of 4.3–6.5%, (Karl et al., 2019a).

However, little is known about the impact of ship emissions in harbour cities of the North & Baltic Sea region. Even if emissions of in-port ships account for only a few percentage of the global emissions related to shipping (Dalsøren et al., 2009), they can have an important impact on local AQ in harbour cities, due to additional emissions from manoeuvring, mooring and diesel powered activities at berth, such as lighting, cooling, heating and sanitation (Meyer et al., 2008). Viana et al. (2014) performed a literature review with the aim of characterising and quantifying the contribution of the maritime

transport sector to air quality degradation along European coastal areas. The reviewed studies agreed on the relevance of ship emissions in coastal areas for PM, NO$_x$ and SO$_2$ and identified a large spatial variability, with maximal contributions in the Mediterranean basin and the North Sea. On average, shipping emissions in the coastal North Sea region contribute with 7–24% to NO$_2$ annual mean and 3–5% to PM$_{2.5}$ annual mean concentrations in the North Sea, while in the Mediterranean PM$_{2.5}$ from shipping contributes with 4–20% (Viana et al., 2014).

Only few studies investigated the impact of in-port ship emissions on the AQ in harbour cities of the Baltic Sea. Saxe and Larsen (2004) showed the impact of local shipping activities in Copenhagen, Denmark, which connects the ship traffic between North and Baltic Sea. NO$_x$ from shipping was exceeding 200 µg m$^{-3}$ of NO$_x$ and causing values of 50–200 µg m$^{-3}$ over several square kilometres of central Copenhagen, while PM and SO$_2$ contributed with insignificant mass concentrations of PM in populated areas near the harbour (Saxe and Larsen, 2004). Pirjola et al. (2013) measured particulate and gaseous

emissions from ship diesel engines with different after-treatment systems using a mobile laboratory inside the harbour areas in Helsinki and along the narrow shipping channel near Turku, Finland, and concluded the need for additional regulation of shipping particulate emissions beyond controlling the fuel sulphur content. Also in Helsinki, Soares et al. (2014) investigated the impact of emissions from ship traffic in the harbours of Helsinki and in the surrounding area on concentrations and exposure identifying a contribution of about 3% to PM$_{2.5}$ concentrations by shipping activities.

A more recent study by Ledoux et al. (2018) in the North Sea port of Calais showed the direct influence of in-port shipping to $SO_2$, $NO_2$ and $PM_{10}$ average concentrations with 51%, 15% and 2% respectively, with substantial concentration peaks synchronized with departures and arrivals of ferries. In the harbour city Hamburg, Ramacher et al. (2018) identified maximum relative contributions from shipping to total $NO_2$ and $PM_{2.5}$ concentrations with 23% and 3% in January and 45% and 16% in July 2012 with highest concentrations located in the port area of Hamburg. A study in preparation (Tang et al. 2019) modelled local $NO_2$ shipping contributions to air pollution in the urban area of Gothenburg of about 14% and a regional $NO_2$ contribution of up to 41% on average to the annual mean, indicating the same importance in controlling local shipping emissions as e.g. road traffic emissions, while $SO_2$ and $PM_{2.5}$ contributions are negligible.

Exposure to air pollution can lead to asthma, respiratory and cardiovascular diseases, lung cancer and premature deaths according to the World Health Organization (WHO, 2006). Corbett et al. (2007) showed that shipping-related PM emissions are responsible for approximately 60,000 cardiopulmonary and lung cancer deaths annually, with most deaths occurring in coastal regions of Europe, East Asia, and South Asia. An update of this study shows that despite implemented regulations, low-sulphur marine fuels will account for 250,000 deaths annually in 2020 due to increase in transport by sea (Sofiev et al., 2018b). Approximately 230 million people are directly exposed to these shipping emissions in the top 100 world ports (Merk, 2014). The large majority (95%) of Europeans living in urban environments are exposed to levels of air pollution considered dangerous to human health. The average contribution of shipping emissions to the population exposure from primary $PM_{2.5}$, $NO_x$, and SOx is 8%, 16.5%, and 11%, respectively, across Europe (Andersson et al., 2009). While exposure to $PM_{2.5}$ was estimated to be a leading cause of WHO environmental burden of disease in six selected European countries (Hänninen et al., 2014), the relationship between $NO_2$ and health is scientifically not as well founded as for $PM_{2.5}$ (WHO, 2006; Heroux et al., 2013). However, $NO_2$ is usually regarded as an indicator of other pollutants and long-term residential exposure to $NO_x$ is moving into focus due to rising evidence for severe health-effects of the respiratory system (WHO, 2016; Wing et al., 2018; Hamra et al., 2015) and as risk factor for myocardial infarction (Rasche et al., 2018). In terms of exposure to shipping emissions, $NO_2$ was found consistently associated with total non-accidental mortality and specific cardiovascular mortality in the Baltic Sea harbour city Gothenburg (Stockfelt et al., 2015). Thus, exposure to air pollution caused by shipping activities in harbour cities needs to be reduced and emissions regulated.

Regulations for the prevention of air pollution from ships was introduced in the Marine Pollution Convention (MARPOL) Annex VI by the IMO and entered into force in 2005. Many countries have ratified this protocol particularly for limiting $NO_x$ and $SO_2$ emissions from ships. The coastal areas of the North Sea and the Baltic Sea have been classified as Sulphur Emission Control Areas (SECA), where the sulphur content in marine fuels is limited to 0.1% from 2015 on. Moreover, the European Union has introduced a requirement limiting the sulfur content in fuels used by ships at berth to 0.1% in 2010. The European Environment Agency (EEA) therefore estimated the decrease of $SO_2$ ship emissions to be 54% between 2000 and 2010 and a further decrease is expected from 2020 onwards due to changes in technology and global regulations (EEA, 2013). It is also expected that this will lead to a decrease in emissions of $PM_{2.5}$. Nevertheless, $NO_x$ emissions from international maritime transport in European waters are projected to increase and could be equal to land-based sources by 2020. In order to

reduce $NO_x$ emissions from shipping, a $NO_x$ Emission Control Area (NECA) will be implemented in the North and Baltic Seas on 1 January 2021. The goal is to decrease nitrogen oxide emissions from maritime transport by 80% compared to present levels on the long run. Besides this, an additional reduction in $PM_{2.5}$ is expected in the future due to less $NO_x$ induced secondary organic aerosol (SOA) formation, which lowers the ship-related $PM_{2.5}$ by 72% in 2040, compared to present-day, while it is

reduced by only 48% without implementation of the NECA (Karl et al., 2018). Despite these regulations to reduce SOx (SECA) and $NO_x$ (NECA) emissions in Europe, ship traffic is still the least regulated sector in Europe compared to other types of anthropogenic emission sources such as road traffic, industrial sources, power generation, or residential heating. Hence, shipping emissions are increasing in terms of the relative weight of shipping emissions to the total of anthropogenic emissions on the regional and local scale in Europe (EEA, 2013). Taking into account the projected increase of maritime transport due

to growth of global-scale trade (Lloyds Register Marine, 2014, EC 2012) as well as the simultaneous increase in population growth and urbanization in coastal areas (Neumann et al., 2015) it is necessary to come up with pollution prevention efforts for ports in harbour cities.

The objective of this study is to identify the impact of emissions due to local shipping activities on air quality and population exposure to concentrations of $NO_x$ in three major Baltic Sea harbour cities: Rostock (Germany), Riga (Latvia) and

the urban agglomeration of Gdansk-Gdynia (Poland). To identify the impact of local shipping activities on AQ, an urban-scale chemistry transport modelling (CTM) system, was set up for the selected Baltic Sea harbour cities. Besides city-specific emission inventories for land-based emission sources, spatially and temporally high-resolution shipping emission inventories have been modelled and applied. All study areas are located in the SECA and the study was performed for 2012 conditions, when the sulphur content in marine fuels was limited to 1% in the region and 0.1% for ships at berth. Therefore, and because

of the decreasing importance, we excluded $SO_2$ from the study focus. We analysed concentrations of $NO_2$, $O_3$ and $PM_{2.5}$ for 2012 conditions and evaluated these with local measurement network data of each harbour city. The impact of local shipping activities on urban air quality has been determined with the perturbation method (zero-out scenario runs). We focus on the impact of local in-port shipping on the air quality in harbour cities, while considering the influence of ocean-going shipping on the Baltic Sea is beyond the scope of this study. Based on the simulated concentration fields, dynamic population weighted

outdoor exposure to $NO_2$ and $PM_{2.5}$ for all urban domains was calculated in different microenvironments using a newly developed generic exposure modelling approach based on publicly available data. This study mainly focuses on $NO_2$ exposure, taking into account the high contributions of local shipping activities to $NO_2$ in other harbour cities, the growing importance of $NO_2$ as indicator for health effects and the usage of $NO_2$ as indicator for health effects due to other pollutants.

To our knowledge, this study is the first one investigating the impact of emissions from local shipping activities on

air pollutant concentrations and population exposure in Baltic Sea harbour cities since the 2010 commenced 0.1% sulphur fuel requirement in harbours (European Parliament Directive2005/33/EC), using a CTM system with high spatial and temporal resolution.

Section 2 of this paper describes the model & data setup, introducing the urban-scale CTM EPISODE-CityChem in Sect. 2.1 and describing the setup of each urban domain in Sect. 2.2 and Sect. 2.3. This is followed by the description of local

emission inventories and their application in the CTM system (Sect. 2.4 and Sect. 2.5). Finally, a new generic approach to achieve outdoor exposure for different microenvironments will be introduced in Sect. 2.7. In Sect. 3, the simulated concentrations will be evaluated (Sect. 3.1) and total as well as ship-related concentration distributions of $NO_2$ and $PM_{2.5}$ will be presented for the city domains (Sect. 3.2). This is followed by the analysis and illustration of exposure results due to total and ship-related concentrations (Sect. 3.3). Section 4 discusses the exposure results with respect to the novel approach for generic dynamic population activity and is followed by conclusions in section 5.

## 2 CTM & Exposure simulation setup

A CTM system with the EPISODE-CityChem model (Karl et al., 2019b; Karl and Ramacher, 2018) to simulate present day urban concentrations of $NO_2$ and $PM_{2.5}$ as well as the contribution of shipping activities to urban air quality was setup for the Baltic Sea urban areas of Rostock, Riga and Gdansk-Gdynia. City-specific meteorological fields, regional boundary conditions, land-based emission and shipping emission inventories have been gathered and modelled. The contribution of present shipping emissions to the modelled concentration of air pollutants was determined from the difference between 'base' runs, which include all emissions, and 'no ship' runs, which exclude emissions from ship traffic (zero-out method). The concentration results are then evaluated and used to model dynamic population-level exposure in different microenvironments for each city.

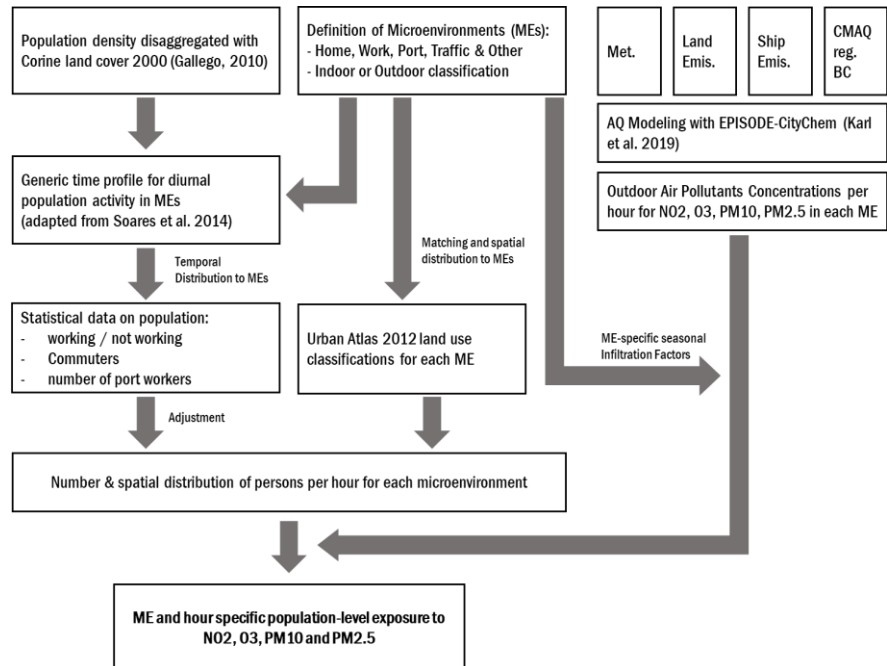

**Figure 1: Study design to calculate microenvironment-specific population exposure to outdoor air pollution based on CTM concentration simulations and taking into account seasonally changing infiltration factors for indoor environments.**

## 2.1 EPISODE-CityChem

For all harbour cities, the urban-scale CTM EPISODE-CityChem (Karl et al., 2019b) was applied. The city-scale Chemistry (CityChem) model is an extension of the urban dispersion model EPISODE of the Norwegian Institute for Air Research (NILU) (Slørdal et al., 2003; Slørdal et al., 2008). A more up-to-date description of EPISODE is in preparation (Hamer et al., 2019). EPISODE systematically combines a 3-D Eulerian grid model with a sub-grid Gaussian dispersion model, allowing for the computation of pollutant concentrations near road traffic line sources and industrial point sources with high spatial resolution. EPISODE-CityChem is capable of modelling the photochemical transformation of multiple pollutants along with atmospheric diffusion to produce pollutant concentration fields for an entire city on a horizontal resolution of 100 m or even finer. The purpose of EPISODE-CityChem is to fill the gap between regional-scale air quality simulations with Eulerian CTM systems (with typical resolutions between 100 m and 1000 m) on one side and micro-scale simulations of limited areas of the urban environment using large eddy simulation (LES) techniques (Nieuwstadt and Meeder, 1997), on the other side. In order to resolve chemical transformation of reactive pollutants in proximity of emission source objects (point source and lines sources), the atmospheric chemistry is considered in detail within the Eulerian grid and in a simplified manner for the sub-grid dispersion. The applied chemical scheme in this study is the EmChem03-mod which is an update of the EMEP45 chemical mechanism (Simpson et al., 2003; Walker et al., 2003), and consists of 45 gas-phase species, 51 thermal reactions and 16 photolysis reactions. Levels of $PM_{2.5}$ and $PM_{10}$ in the model are controlled by primary emissions of particulate matter and their atmospheric dispersion, while secondary aerosol formation is not considered in the model (Karl et al., 2019b).

The model reads meteorological fields either generated by the prognostic meteorology component of the Australian air quality model TAPM (The Air Pollution Model; Hurley, 2008; Hurley et al., 2005) or other diagnostic wind fields, for calculating the dispersion parameters, vertical profile functions in the surface layer, and the eddy diffusivities in EPISODE-CityChem. Moreover, EPISODE-CityChem has the option to use the time-varying 3-D concentration field at the lateral and vertical boundaries from the Community Multiscale Air Quality Modelling System (CMAQ, Byun and Schere, 2006) as initial and boundary concentrations for selected chemical species.

Emissions in EPISODE-CityChem can be treated as area sources (2-dim. area of the size of a grid cell), line sources (line between two (x, y)-coordinates), and point sources (industrial and power plant stacks). Moreover, a simplified street canyon model (SSCM) based on the OSPM model (Berkowicz et al., 1997) can be used in EPISODE-CityChem, potentially allowing for a better treatment of $NO_x$ at traffic stations. The Meteorological Pre-Processor (WMPP) of the Weak-wind Open Road Model (WORM, Walker, 2011) is used in the point source sub-grid model to calculate the wind speed at plume height for the dispersion of plume segments released from industrial and power plant stacks.

Emission input containing sector-specific (following SNAP nomenclature) and geo-referenced yearly emission totals are pre-processed with the model's interface for emission pre-processing, the Urban Emission Conversion Tool (UECT, Hamer et al., 2019), which produces hourly varying emission input for point sources, line sources and area source categories using sector specific temporal profiles and vertical profiles.

In this study, we defined three urban domains for CTM simulations with EPISODE-CityChem (Figure 2). EPISODE-CityChem uses a 2-D Cartesian coordinate system and therefore, we used the Universal Transverse Mercator (UTM) conformal projection to set the geographic dimensions for all research domains. While the model domains for Rostock and Riga were set-up for a $16 \times 16$ km² and a $20 \times 20$ km² area with 400 m resolution, the model domain for the Gdansk-Gdynia urban agglomeration was set-up for a $40 \times 40$ km² area with 1 km grid resolution. The SSCM for traffic line sources was activated for all simulations and EPISODE-CityChem provided concentration output and other diagnostic output in netCDF files.

## 2.2 Meteorology setup

In this study, the meteorological data for all research domains was provided from the meteorological component of the coupled meteorological and chemistry transport model TAPM. TAPM predicts three-dimensional meteorology based on an incompressible, non-hydrostatic, primitive equation model with a terrain-following vertical coordinate for three-dimensional simulations. The model solves the momentum equations for horizontal wind components, the incompressible continuity equation for vertical velocity, and scalar equations for potential virtual temperature and specific humidity, cloud water/ice, rain water and snow (Hurley, 2008). A vegetative canopy, soil scheme, and urban scheme are used at the surface, while radiative fluxes, both at the surface and at upper levels, are also included. TAPM includes a nested approach for meteorology, which allows a user to zoom-in to a local region of interest quite rapidly, while the outer boundaries of the grid are driven by synoptic-scale analyses.

In this study, three-hourly synoptic scale ECMWF ERA5 reanalysis ensemble means on a longitude/latitude grid at 0.3 degree grid spacing have been used to drive the meteorological module of TAPM for all urban domains. Moreover, land cover classes and elevation have been updated with Corine Land Cover 2012 data (CLC2012, Copernicus Land Monitoring Service, 2012) and the Digital Elevation Model over Europe (EU-DEM, EEA, 2017) to account for urban-specific features. For each city, multiple nested meteorological domains have been set up (Figure 2), to simulate meteorological fields with hourly values in year 2012.

## 2.3 Boundary conditions

The boundary conditions as concentration values at the lateral and vertical boundaries of the urban domains in EPISODE-CityChem are based on results from regional model simulations in the North and Baltic Sea performed for the year 2012. The regional simulations have been performed with CMAQ on a grid resolution of $4 \times 4$ km² and a temporal resolution of one hour (Karl et al., 2018). CMAQ model simulations were driven by the meteorological fields of the COSMO-CLM (Rockel et al., 2008) version 5.0 using the ERA-Interim re-analysis as forcing data. The meteorological runs were performed on a $0.11° \times 0.11°$ rotated lat-lon grid using 40 vertical layers up to 20 hPa for entire Europe. High-resolution meteorology obtained from COSMO-CLM on a $0.025° \times 0.025°$ grid resolution was used for the North Sea and the Baltic Sea regional simulations with CMAQ. Chemical boundary conditions for the model simulations were provided through hemispheric CTM simulations, from a SILAM model run on a global domain with $0.5° \times 0.5°$ grid resolution, which was provided by Finnish

Meteorological Institute (Sofiev et al., 2018a). Land based emissions for the model simulations were calculated at Helmholtz-Zentrum Geesthacht (HZG) with the SMOKE for Europe (SMOKE-EU) emission model (Bieser et al., 2010; Backes et al., 2016), version 2.4. The regional concentrations of simulations with and without shipping emissions were evaluated against measurements and showed strong underestimations of $PM_{2.5}$ (regionally by up to -70%) in summer by CMAQ (Karl et al.,
2019a). After evaluation, the regional concentrations were interpolated to the specific resolutions of each urban domain, applied at the lateral boundaries in EPSIODE-CityChem and used to simulate 2012 hourly concentrations of $PM_{2.5}$ and $NO_2$. The same regional CTM system was used in a study in preparation (Tang et al., 2019) to perform local CTM simulations in the Gothenburg area with the chemistry transport module of TAPM but with a different preparation of boundary concentrations from CMAQ: TAPM allows just 1-d boundary concentration fields with time being the only variable, and therefore the TAPM
boundary concentrations were calculated using horizontal wind components on each of the four lateral boundaries for weighting the boundary concentrations.

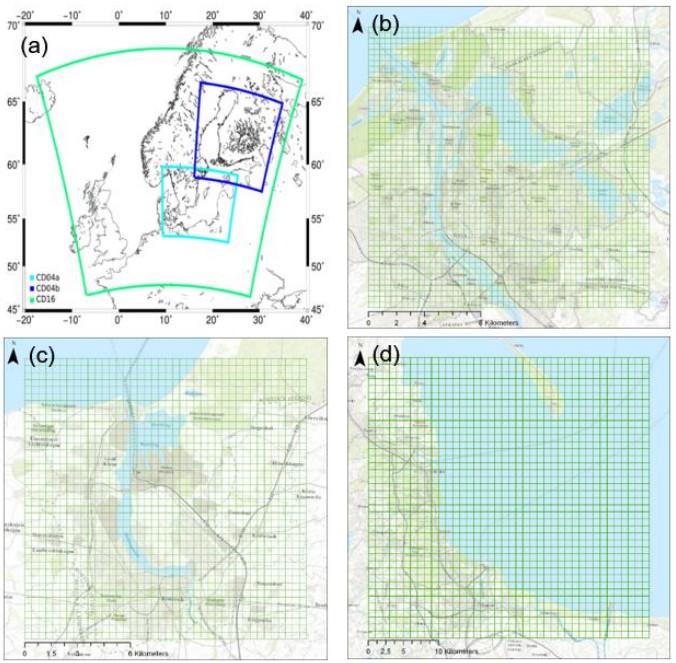

**Figure 2: (a) Regional CTM simulation domains, which have been used to drive the local-scale EPISODE-CityChem simulation for**
**the urban domains in (b) Riga and (c) Rostock with 400 m resolution for $20 \times 20$ km² and $16 \times 16$ km² extent, and (d) Gdansk-Gdynia with 1000 m resolution and $40 \times 40$ km² extent.**

**2.4 Land-based local emission inventories**

        Matthias et al. (2018) have discussed the necessity to utilize emission data in high spatial and temporal resolution on a coordinate grid that is in agreement with the CTM grid, due to emission data being probably the most important input for
chemistry transport model (CTM) systems. Therefore, we account for local land-based emissions in every sector based with

city-specific or downscaled emission data from regional emission inventories if city-specific data was not available. Subsequently, the annual totals were applied in the UECT interface for EPISODE-CityChem to produce hourly emissions for area, line and point source emission categories. The following describes the compilation of the emissions for the three source types (point, line and area sources).

The line source category was assigned to road or rail transport emissions only. For the Rostock domain, the traffic emissions have been provided by the German Federal Environmental Agency (Schneider et al., 2016) as gridded area annual emission totals with a resolution of 400 m. These gridded emissions were redistributed to the major road network based on Open Street Map road types and weighted by traffic activity with FME® (Feature Manipulation Engine) which is an ETL (Extract Transform Load) software for GIS data. First, OSM road types (Trunk and motorway, primary and secondary, tertiary)

were matched with the corresponding traffic categories (highway, rural, urban) as established in the Deutsches Zentrum für Luft- und Raumfahrt (DLR) traffic emission project 'Verkehrsentwicklung und Umwelt' (VEU, Seum et al., 2015). Second, the VEU data was inspected to identify the ratio of total annual German traffic emissions for each traffic category. Third, the identified ratio was used to distribute the gridded traffic emissions to OSM roads and a total of 3,875 traffic line sources were obtained. For Riga, the environmental service company Estonian, Latvian and Lithuanian Environment (ELLE), has provided

annual total traffic emission data, including railway line sources as well as regular ferry lines. The regular ferry lines were excluded because they are covered in the shipping emission inventory separately. Emission data for line sources by ELLE referred to the year 2014 and was used for 2012 without scaling. A total of 2,875 line source objects were included in the calculations. For the urban agglomeration of Gdansk-Gdynia emissions from vehicular traffic were provided as line sources by ARMAAG, the air quality monitoring organization of Gdansk. A total of 9,884 line source objects were included in the

calculations.

The point source category applied to industrial facilities and power plants as listed in the available datasets. In the Rostock domain, also small energy production and commercial combustion sources within the municipality of Rostock were represented as point sources. Data on annual total emissions as well as stack-specific characteristics, such as emission height, exit velocity and temperature, were provided by the Department for Environment, Nature protection and Geology (LUNG) of

the federal state Mecklenburg-Vorpommern. A total of 32 point sources were allocated to the city domain of Rostock. In Riga and Gdansk-Gdynia, again energy production and commercial combustion sources in the urban area were represented as point sources. Data on point sources emissions in Riga was provided by ELLE and in Gdansk-Gdynia by ARMAAG. Additional to the total annual emissions, stack characteristics for 719 point sources in Riga and 676 point sources were estimated based on the data set on European stacks and associated plume rise published in Pregger and Friedrich (2009).

The area source category was used for the remaining emission categories, such as domestic heating, agricultural emissions and solvent use. For Rostock, domestic heating, solvent use and agricultural emissions were provided as gridded emissions with $400 m^2$ resolution by the German Federal Environmental Agency (Schneider et al., 2016). For Riga and Gdansk-Gdynia, annual total emissions of the same categories were extracted from the SMOKE-EU emission dataset. The SMOKE-EU area emissions with a resolution of 5000 m were downscaled to 400 m grid-resolution for Riga and 1000 m for Gdansk-

Gdynia respectively. The downscaling utilized CLC2012 land use information and a population density grid of the European Union (Gallego, 2010) as proxy data.

The collected total annual land-based emission inventories for each urban domain were then distributed over time in UECT (see sect. 2.1) for each sector by temporal disaggregation using sector-specific monthly, weekly and hourly profiles
(adopted from SMOKE-EU).

## 2.5 STEAM ship emissions

The Ship Traffic Emission Assessment Model (STEAM, (Jalkanen et al., 2009; Jalkanen et al., 2012; Johansson et al., 2013; Johansson et al., 2017) was used to create shipping emission inventories for Rostock, Riga and Gdansk-Gdynia. Automatic Identification System (AIS) data from the Baltic Sea countries were used in this work together with the technical
description of the global fleet (IHS, 2017). The emissions from ships in port areas were provided in two height layers, below 36 m and above it, to account for stack height differences between various types and sizes of ships. For Rostock, hourly gridded emissions on 250 m resolution for the port of Rostock and parts of the Baltic Sea within the model domain were provided by FMI with the STEAM model, based on AIS records in 2012. The ship emissions were interpolated to 400 m grid resolution for the use as area sources in EPISODE-CityChem. Area emissions from shipping representing moving ships were distributed
vertically equally over the lowest four model layers of EPISODE-CityChem (each layer having 25% of the total area emission) covering a vertical profile up to 87.5 m height above sea-level. For Riga and Gdansk-Gdynia the same approach was used: Gridded emissions on 250 m resolution for the ports and parts bays inside the model domain of Riga and Gdansk-Gdynia were provided by the STEAM model and interpolated to area sources with 400 m and 1000 m grid resolution, respectively. A challenge for port emission inventories is that energy usage of various kinds of ships is often unknown, which may lead to
significant uncertainties concerning predictions of auxiliary engines and boiler fuel consumption and emissions. These are often estimated based on vessel boarding programs (Hulskotte and van der Denier Gon, 2010; Starcrest Consulting Group, LLC, 2014) or determined from vessel cargo capacity (Jalkanen et al., 2012; Johansson et al., 2013). Several models for vessel propulsion power predictions as a function of speed exists, but relatively little is known about power profiles of auxiliary systems during port stays.

## 2.6 Generic population-level exposure modelling

### 2.6.1 Population-level exposure modelling

Population exposure estimates are used in epidemiological studies to evaluate health risks associated with impacts of air pollution on human health. While the principle idea of exposure is the pollutant concentration values in the environments
where people spend their time, and the amount of time they spend within them (WHO, 2006), there exist several modelling approaches for this principle idea. Özkaynak et al. (2013) ranked exposure metrics relevant to air pollution epidemiology studies by their complexity: Beginning with (1) measurements of concentrations at monitoring sites as simplest exposure

metric, over (2) land-use regression modelling of concentrations, followed by (3) AQ modelling with CTM and (4) data blending with satellite data, the most complex metric is (5) exposure modelling. Traditional exposure model approaches assume that concentrations of air pollutants at the residential address of the study population are representative for overall exposure (Ott, 1982). Since Ott (1982), this approach is known to introduce potential bias in the quantification of human health

effects, as the individual and population-level mobility is not accounted for. Nevertheless, state-of-the-art exposure modelling studies have overcome this traditional approach and are using population activity data and models, to account for the diurnal variation of population numbers in different locations (e.g. Reis et al., 2018; Bell, 2006; Xu et al., 2019; Beckx et al., 2009; Beevers et al., 2013; Soares et al., 2014). Thus, to model population numbers suitable for exposure calculations, it is generally necessary to know the population distribution and characterization and therefore the number of people and diurnal activity

patterns of different characteristic population groups. While annual gridded population numbers in different spatial resolutions and other annual population characteristics such as age distribution or status of employment are available in publicly available databases for many countries in the world, profiles on average time spent daily in a specific environment are mostly the subject of national or municipal surveys and are scarce. Moreover, surveys have shortcomings such as lacking representativeness and therefore oversimplification of social reality. Recent population activity based exposure studies focus on utilising mobile

devices to assess mobility (Jiang et al., 2012; Picornell et al; Dewulf et al., 2016; Nyhan et al., 2016; Glasgow et al., 2016) . Nevertheless, the number of studies published with such data is limited up to now because of data protection and privacy issues and problems accessing the data (Ahas et al., 2010) and the outcomes mostly describe individual activity patterns which need to be up-scaled to population level exposure. A link between individual and population level exposure is the concept of Microenvironments (MEs), which is defined by a location or area in which human exposure takes place, containing a relatively

uniform concentration, such as, e.g. home or workplace. Therefore, MEs allow for clustering individual exposure to population level exposure in an area where the air pollutant concentrations can be assumed to be homogenous. Moreover, the concept of MEs allows for the consideration of outdoor air pollution infiltrating into different indoor environments (Borrego et al., 2009). This is necessary because people spend most of their time indoors in buildings. To reduce outdoor air pollution entering indoor environments, modern buildings can be equipped with air intake filters with different efficiencies, depending on their size,

technique and position (SeppĘnen, 2008). Hence, when evaluating human exposure it is essential to estimate the concentrations of the air pollutants not only in open air, but also in different indoor locations (Leung, 2015; Schweizer et al., 2007; Sørensen et al., 2005; Baek et al., 1997). Outdoor locations that can exhibit similar air pollutant concentrations can also be termed MEs.

Besides these challenges in modelling population activity for population level exposure estimates, atmospheric chemistry transport models, as applied in this study, can provide consistent spatio-temporal air pollution concentration fields

for exposure assessments. With the established AQ model system in this study it is possible to calculate concentration fields with hourly concentration values, which represent an area of 100 x 100 m², but it is still necessary to model the population distribution within Baltic Sea harbour cities with the same temporal and spatial resolution. Therefore, we developed a generic approach to model population activity in different MEs of Baltic Sea harbour cities using the Copernicus Urban Atlas 2012 land use and land cover data in combination with literature-based, generic and microenvironment specific, diurnal activity

data, under consideration of indoor and outdoor environments. The product of this generic approach is a set of maps with numbers of citizens in different microenvironments and hours of the day. These maps can then be used to calculate population-level outdoor exposure using consistent spatio-temporal air pollution concentration fields.

### 2.6.2 Generic modelling of human activities

5       To derive temporally and spatially disaggregated population activity in Rostock, Riga and Gdansk-Gdynia we created and followed the following four steps. First, we separated the population activity into five different microenvironments (MEs): home environment (ME_home), work environment (ME_work), port work environment (ME_port), road traffic environment (ME_traffic) and other outdoor environment (ME_other). In a second step, we mapped these MEs to suitable Copernicus European Urban Atlas 2012 (UA2012) classifications (https://land.copernicus.eu/local/urban-atlas) of urban land use for the
spatial aggregation of MEs (Copernicus Land Monitoring Service, 2016). Table 2 shows the result of mapping MEs to UA2012 categories. For a detailed description of all UA2012 classifications provided by Copernicus, see supplement I. The UA2012 land use classifications are the result of satellite imagery. Therefore, it is often not possible to differentiate building structures in dense urban areas into residential or commercial buildings, but it is possible to identify e.g. roads, industrial areas, port areas, green areas or water bodies. Accordingly, we made assumptions to allocate ME_home and ME_work with 30% and
70% to the "continuous dense urban fabric" class in UA2012 to take into account commercial activities and offices in more dense urban areas. Thus, ME_home shall represent the population residencies of all citizens in the research domain, while ME_work shall represent workplace addresses and ME_port designated port areas in every urban research domain. Moreover, the ME_traffic is limited to the road network, whereas rail, shipborne and aviation transport modes are neglected because of uncertainties associated with the classifications of respective attributed land use areas. The areas in the UA2012 relating to the
excluded transport modes often include associated land and therefore huge areas, which are not accessible for people in transit. ME_other is mapped to sports and leisure facilities, as well as green urban areas and is therefore representing outdoor activities such as sports and outdoor recreational activities. However, indoor activities were not integrated in ME_other, because the information could not be extracted from UA2012. Nevertheless, we classified the MEs as indoor or outdoor environment (Table 2) to consider outdoor pollution infiltrating indoor environments. For the indoor environment ME_home and ME_work
we used infiltration factors (IF) in the calculation of exposure to ambient air pollution concentrations of $NO_2$ and $PM_{2.5}$, which we derived from Borrego et al. (2009) and which are based on Baek et al. (1997), Chau et al. (2002) and Dimitroulopoulou et al. (2006). No specific analysis of the availability of air intake filters in the research domains Rostock, Riga and Gdansk-Gdynia was done.

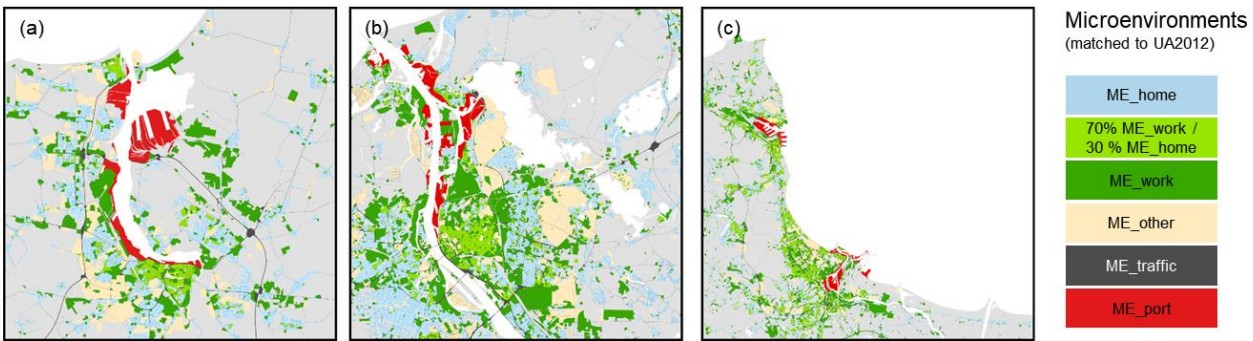

**Figure 3: Urban Atlas Land use classifications, aggregated by colours according to microenvironment mapping presented in Table 2 for Rostock (a), Riga (b) and Gdansk-Gdynia (c).**

The third step was the calculation of static population taking into account city-specific statistics. Static population was calculated with raster data on population density using the Copernicus Corine Land Cover (CLC) inventory with values corresponding to density in inhabitants per square kilometre (Gallego, 2010). The advantages of this approach are (1) a unified approach to estimate population in the total research domain and (2) the consideration of suburban and rural areas which do not only take into account the city's population but the entire domain of interest. Besides, a comparison of population derived from the population density grid shows good agreement with municipality population statistics of each city (Table 3), with slightly higher values for the region due to residencies surrounding the city limits.

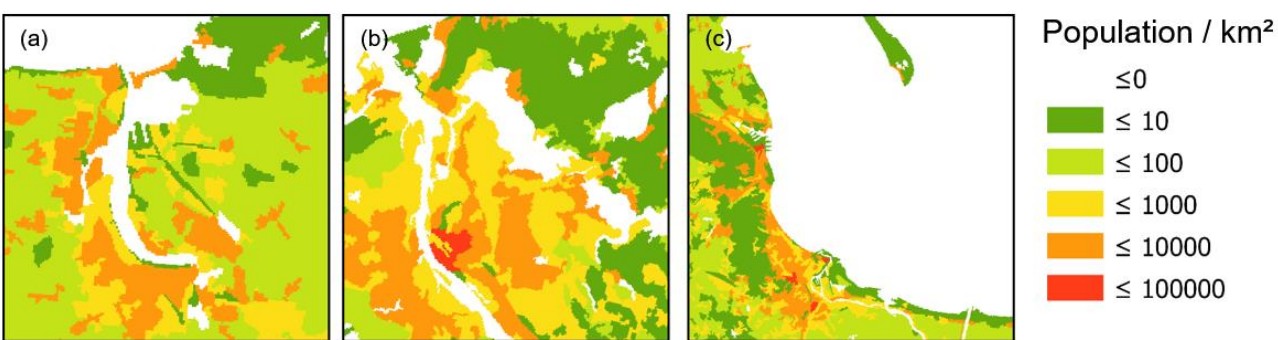

**Figure 4: Population Density per km² as derived from (Gallego, 2010) in (a) Rostock, (b) Riga, and (c) Gdansk-Gdynia.**

In a fourth step, we assembled generic diurnal variation of population activity for each ME to temporally distribute the population to all MEs because there exists no specific information for Rostock, Riga or the Gdansk-Gdynia area. The generic time profiles are mainly derived from diurnal variation of population activity in the Helsinki Metropolitan Area in four MEs: home, workplace, traffic and other (Kousa et al., 2002; Soares et al., 2014). Soares et al. (2014) derived information on

Helsinki population from annually collected data of the municipalities of the Helsinki Metropolitan Area. We compared these with other diurnal activity patterns in Europe (Brook and King, 2017; Borrego et al., 2009) and figured out similar diurnal patterns, such as a high amount of people in the home environment during night, a growing number of people working during the day with a peak around noon followed by a decrease until early evening and traffic rush hours in the morning and evening.

Therefore, we consider the adapted pattern shown in Figure 5 to be suitable for other Baltic Sea harbour cities. Nevertheless, we analysed the relation of employed people and the daily maximum of work activity in Helsinki to assimilate the daily maximum work activity in the generic profile for each city, to account for dynamics in the second largest ME (ME_work) and scaled all other MEs uniformly.

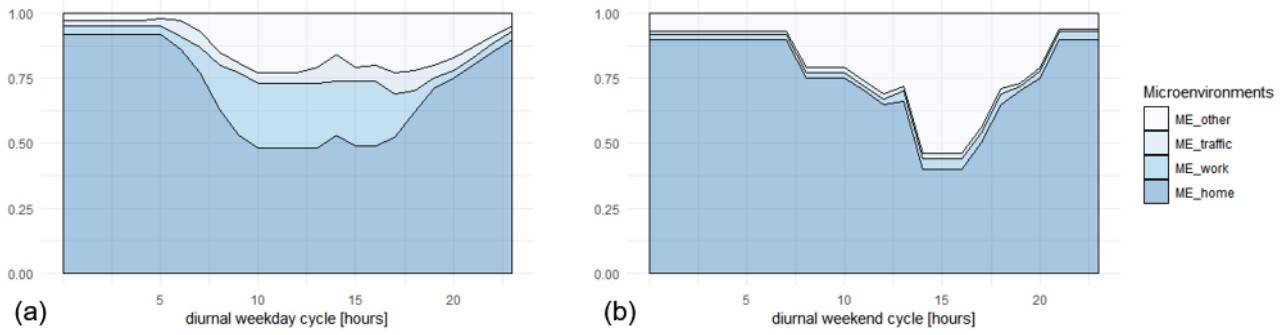

**Figure 5: Generic diurnal activity patterns during weekdays (a) and weekends (b), adapted from Soares et al. (2014).**

While we use this generic profile for weekdays, we additionally adapted a weekend profile with less work and higher other activities from the study by (Borrego et al., 2009) to account for daily patterns (Figure 5) but we did not account for

holidays. Another consideration is the integration of daily commuters during workdays. We gathered data on commuting rates from the municipality of each city and assigned the total number of commuters to ME_traffic in morning/evening rush hours and ME_work during the day. When it comes to population working in the ME_port, we assigned port work as part of the ME_work but with detailed numbers on workers in the port areas of Rostock, Riga and Gdansk-Gdynia gathered from port-specific statistics. Therefore, we differentiate between numbers of direct port employment and indirect or related port

employment to spatially distribute port workers with the UA2012 port area classification. The UA2012 classification Port Areas is described as the administrative area of inland harbours and seaports as well as infrastructure of port areas, including quays, dockyards, transport and storage areas and associated areas. Thus, it is possible to use the UA2012 port area classification to distribute numbers of workers in direct port employment activities spatially. Moreover, we assumed three-shift operation in the port areas and therefore distributed the harbour workers with 25% to night shift, 50% to day shift (taking

into account administrative work during day) and 25% to late shift. The number of harbour workers is then removed from ME_work.

Following this approach, it is possible to compile the number and spatial distributions of people for every hour of the diurnal cycle and in each defined microenvironment in the form gridded datasets. Therefore, we account for dynamics of a moving population. For this study, we generated created grids with a resolution of 100 m, following the resolution of the simulated concentration fields for $NO_2$ and $PM_{2.5}$.

5 **3 Results**

We evaluate and present results for simulated concentrations in the Baltic Sea harbour cities Rostock, Riga and the urban agglomeration of Gdansk-Gdynia, focusing on $NO_2$. For each city, we performed runs with and without shipping, to determine the effect of local shipping on $NO_2$ concentration levels as well as population-level exposure to $NO_2$. Besides the exposure of all ME due to total concentrations and shipping activities, we analyse the exposure to shipping-related 10 concentrations in ME_home, ME_work and ME_port.

**3.1 Evaluation of simulated concentrations**

Due to an insufficient number of valid time series at the measurement stations in 2012 for Rostock and Riga to achieve significant performance indication, we focus on a discussion of measurement evaluation in the Gdansk-Gdynia agglomeration, which contains eight valid $NO_2$ measurement time series. In Rostock, there are four stations for $NO_2$, while in Riga there are 15 two stations for $NO_2$. However, statistical indicators for $NO_2$, $O_3$ and $PM_{10}$ for all available stations in all cities as well as a detailed description of the AQ simulation performance in Rostock and Riga can be found in Supplement II of this paper.

The analysis of spatial correlations for $NO_2$ time series in Gdansk-Gdynia has shown an r² of 0.3 for station averaged daily averages in 2012 and an r² of 0.79 for station-specific annual averages (Figure 6). The analysis of temporal correlation for hourly values over one year at single stations shows four urban background stations with r values between 0.3 and 0.35 20 and four urban background stations with r values between 0.2 and 0.3. The poorer correlation values can be expected due to not-localised information on temporal emissions. Modelled $NO_2$ for hourly values over one year is in agreement with observed $NO_2$ with overestimation of $NO_2$ at station Wrzescez (urban background station located in an urban green area, Latitude 54.38028, Longitude 18.62028, height asl 40 m) by 4% and underestimation of $NO_2$ (-1% to – 26%) at all other (urban background) stations. $NO_2$ shows overall good performance and FAC2 values for $NO_2$ in Gdansk-Gdynia reach from 0.46-0.7 25 and are therefore fulfilling the acceptance criteria for urban regions of FAC2 $\geq$ 0.3 as defined by Hanna and Chang (2012).

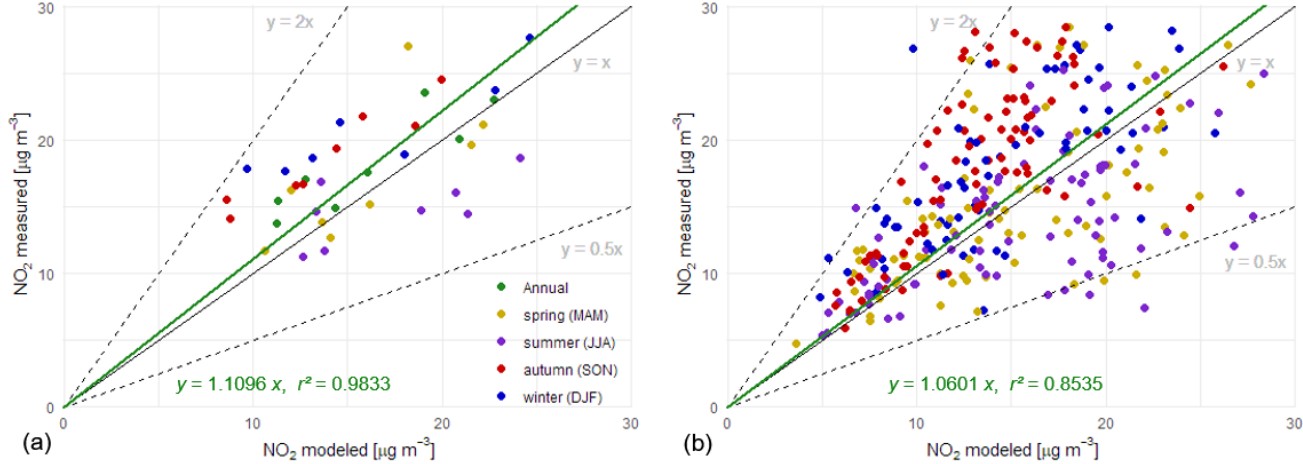

**Figure 6: Modelled versus measured NO₂ concentrations at all available measurement stations in the Gdansk-Gdynia research domain. (a) Shows annual station averages with each dot indicating one station, while (b) shows daily averages with each dot indicating an average of all stations. For (a) and (b) the colours display seasons.**

## 3.2 Predicted concentrations and impact of shipping on NO₂ in 2012

Hourly and annual $NO_2$ concentrations at all available measurement stations throughout 2012 in all harbour cities are mostly below concentration limits as defined by the EU Air Quality Directive: While there are no exceedances for Rostock and Riga, there is only exceedance of the hourly $NO_2$ limit of 200 µg m⁻³ at a station close to the port of Gdansk. The graphical analysis of highest annual mean $NO_2$ concentrations in all urban domains shows three typical areas of elevated $NO_2$ pollution levels above 20 µg m⁻³, which is the guideline value for annual mean concentrations define by WHO (2006); roads with high traffic density, city centres, and port areas as well as areas surrounding the port areas (Figure 7).

The contribution of shipping to the $NO_2$ concentrations (Table 4) in Rostock is significant with 22% impact on $NO_2$ annual averaged grid mean in the complete domain. In Rostock, the shipping impact focuses with high values on areas inside the harbour and decreases rapidly with growing distance to the port areas. For Riga, the contribution of shipping to $NO_2$ concentrations has a lower impact on the total annual averaged grid mean of 11%. It is mainly located along the river Daugava north of the main city but also impacts areas west of the river with concentrations of 3 – 5 µg m-3 $NO_2$. Comparing the spatial patterns of averaged air quality and the impact of shipping in Riga in terms of $NO_2$, it becomes evident, that areas with elevated concentration levels are mostly not overlapping with areas of high $NO_2$ concentrations due to shipping, especially in the city-centre. Thus, shipping is not considered as the main contributor to $NO_2$ concentrations in the city-centre. In Gdansk-Gdynia, the contribution of shipping is low over land. Most of the emissions are transported seawards, leading to enhanced concentration levels in the east and northeast of the most polluted areas, which is not displayed in Figure 7. Due to the main interest in population-level exposure to $NO_2$ concentrations, we show concentrations only in areas with population densities above zero. Nevertheless, the port area of Gdansk, which is located next to the city-centre, shows maximum ship contributions

of up to 20 µg m-3. In total shipping contributes with 16% to the total annual averaged grid mean in the Gdansk-Gdynia domain, whose extent (40 × 40 km²) is four times bigger than for Riga (20 × 20 km²). Although the average contribution of shipping to the total NO2 concentration within the entire modelled domain was modest in all urban research domains, these contributions can be higher than 20% in the vicinity of the harbours within a distance of approximately one kilometre. The total urban area impacted by emissions from shipping, determined as the area with ship-contributed $NO_2$ concentrations above 5 µg m-3, was 5.88 km² for Rostock, 9.26 km² for Riga and 17.42 km² for Gdansk-Gdynia. In relation to the extent of the three study domains, shipping affects an area corresponding to 2.73%, 2.76% and 3.02% of the populated land in Rostock, Riga and Gdansk-Gdynia, respectively.

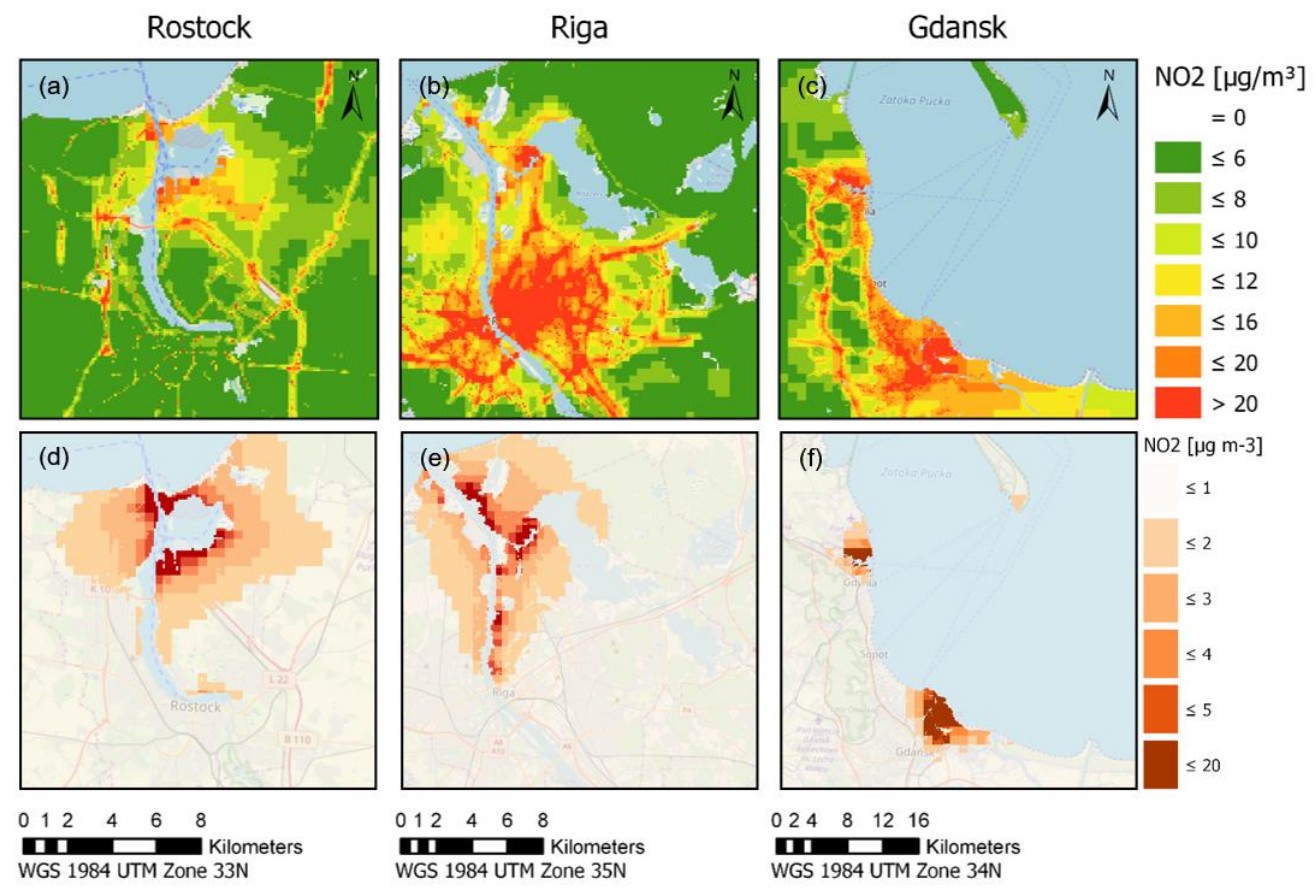

**Figure 7: NO$_2$ annual mean concentrations in Rostock (a), Riga (b) and Gdansk-Gdynia (c), and contribution of local shipping to annual mean NO$_2$ concentration in Rostock (d), Riga (e) and Gdansk-Gdynia (f).**

**3.3 Predicted exposure to NO₂**

**3.3.1 Exposure in all Microenvironments in 2012**

The population level exposure in Rostock, Riga and Gdansk-Gdynia was computed based on the predicted NO₂ concentrations and activities of the population in different MEs. The population data was interpolated on to a rectangular grid with a horizontal grid size of 100 x 100 m², consistent with the pollutant surface concentration grids. The population exposures were computed for each hour of the year, separately for the selected five MEs. Population exposure is a combination of both the concentration and activity (or population density) values. The fractions of exposure to NO₂ in various microenvironments of each urban domain are presented in Figure 8. In all harbour cities, the exposure at home is responsible for most of the exposure, with 59%, 54% and 55% in Rostock, Riga and Gdansk-Gdynia respectively. In Rostock and Gdansk-Gdynia the 2nd highest contributor is the ME_other with 19% and 24%, while in Riga the ME_work comes second with 19%. Nevertheless, in Riga, the ME_other is with 18% almost as high as ME_work. In Rostock and Gdansk-Gdynia, ME_other contributes with 13%. While the ME_traffic in all urban domains is between 7% and 9%, the ME_port is below 1%, indicating a low total exposure in the port areas.

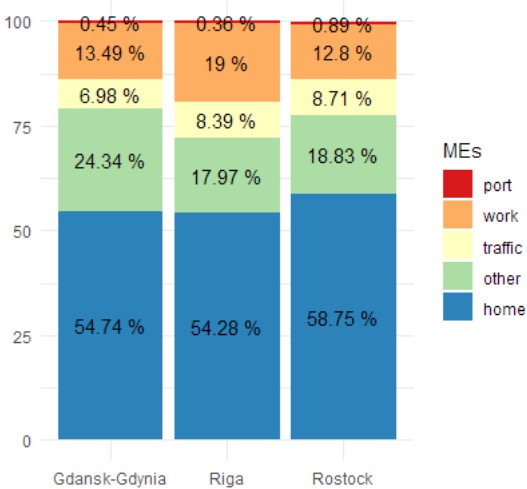

**Figure 8: Relative distribution of total exposure in different microenvironments based on total annual averaged grid mean exposure to NO₂ in Rostock, Riga and Gdansk-Gdynia.**

We have presented the spatial distributions of the predicted annual average population exposures in Rostock, Riga and Gdansk-Gdynia in 2012 in Figure 9 for the total exposure and separately for all microenvironments. These distributions exhibit characteristics of both the corresponding spatial concentration distributions and population activities. There are elevated values in the city centre, along major roads and streets, and in the vicinity of urban district centres. The very high

home and high work exposures in the centre of Riga are caused both by the relatively high concentrations and by the highest population and workplace densities in the area. The spatial distributions of the population exposures at home and work correlate in some regions, especially in the city-centres. This is due to mapping of the UA2012 category "Continuous urban fabric" to ME_home and ME_work, which shall reflect work environments located in the city and district centres, besides workplaces in major industrial, service and commercial regions. Nevertheless, due to less time spent during the day in ME_work, the exposure in ME_home is higher by one order of magnitude. As expected, due to mapping with the UA2012 road classification, the exposure in ME_traffic is limited to the main network of roads and streets, and in their immediate vicinity.

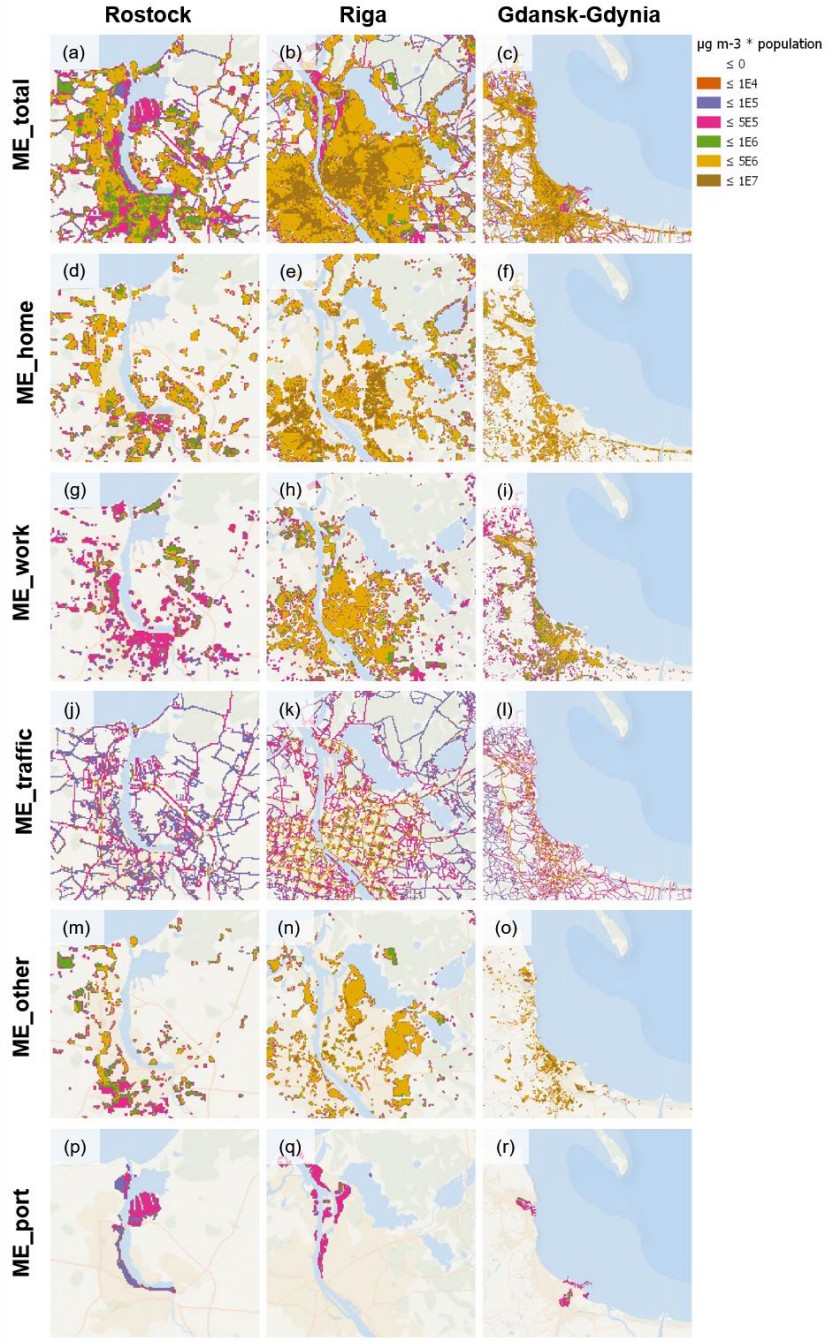

**Figure 9: Exposure to NO₂ from all sources in all microenvironments and urban domains.**

### 3.3.2 Exposures in 2012 due to shipping

To investigate the impact of shipping to total NO2 exposure, we computed the hourly NO2 concentrations due to shipping with the ME-specific population grids of the same spatial and temporal resolution for each urban domain. The contribution of local shipping to the total population exposure as well as to the different MEs to NO2 concentrations in Rostock, Riga and Gdansk-Gdynia are presented in Table 5. Moreover, we have presented in Figure 10 the spatial distributions of annually averaged predicted population exposures to NO2 in Rostock, Riga and Gdansk-Gdynia in 2012, originated from shipping and in the MEs ME_home, ME_work and ME_port.

The population exposure from local shipping in Rostock is responsible for about 13% of the total exposure in all MEs. Thus, shipping is a substantial source of exposure to $NO_2$ in the Rostock urban area. The biggest influence of shipping to $NO_2$ exposure is close to the shore at the port area's exit in the South of the city, which is densely populated and a spot of major attraction in Rostock. In this area, shipping contributes with up to 80% to the annual mean exposure. A detailed analysis of the affected MEs shows a contribution of shipping as total annual averaged grid mean to ME_home which is slightly higher (14%) than the exposure to all MEs. Especially residencies in the North and West of the port areas show high exposure to $NO_2$, again with relative contributions of 80%. The microenvironment with the strongest influence due to shipping is, as expected the ME_port with annually averaged contributions of 46% in the total ME_port. Thus, a reduction of shipping emissions inside the port area, e.g. with onshore power supply, could decrease exposure in the ME_port and therefore the port workers by almost the half with respect to the annual mean. Some areas of the ME_port, especially in the northern parts, the exposure due to shipping is between 50-80% compared to the total exposure from all sources. Regarding the other MEs, the contribution of shipping is about 10-11% as annually averaged grid mean, but for the ME_work also of importance in the northern areas close to the shore. In general, the population exposure caused by shipping is focused in central Rostock, near the main harbours and within some densely inhabited parts of the city and is decreasing in North direction.

In Riga and Gdansk-Gdynia there are similarities to Rostock regarding the decrease of shipping emission related exposure to $NO_2$ with increasing distance from harbour and the importance of residencies close to the port areas. The overall contribution of shipping emissions to the total annual averaged grid mean exposure in all MEs is lower in Riga and Gdansk-Gdynia (5% and 4% respectively). In addition, the annual averaged grid mean contribution of shipping emissions to the ME_port in Riga is similar to Rostock (44%) but lower in Gdansk-Gdynia (26%). Nevertheless, the absolute exposure is in the same order of magnitude in all cities. Thus, besides these gridded means, there are hotspots of the contribution from shipping in some work, port work and residential areas close to the port. In Riga, the entrance to the port and the port itself is located very close to the city-centre and some areas of the ME_work along the river Daugava are substantially exposed to $NO_2$ from shipping, with relative contributions between 40-80%. In the Gdansk-Gdynia study domain, most of the shipping emissions occur outside of the city on the sea. Especially in the port of Gdansk, with its main activities located close to the sea and predominant winds from Southwest, which advect pollutants emitted from shipping away from the city-centre. Nevertheless, the impact of shipping to $NO_2$ exposure is significant close to the harbour and along the coast, especially in the

populated areas in the North of Gdynia but with less relative exposure due to shipping, maximum 60%, compared to exposure from all other sources in Rostock and Riga. Although the coastline of the Gdansk-Gdynia domain shows high absolute exposure to $NO_2$ (Figure 9), shipping only shows impacts of 10-20% near the coastline.

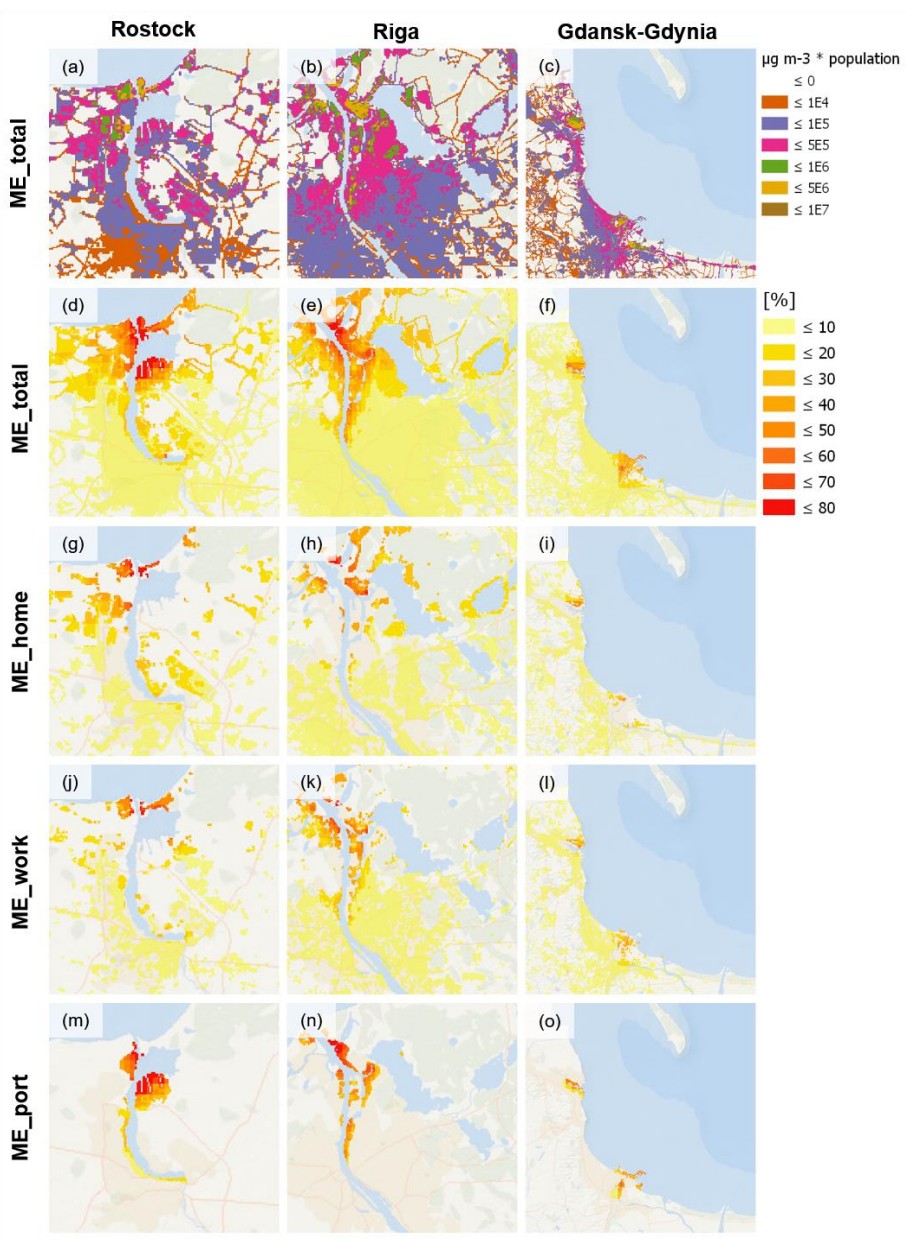

**Figure 10: Exposure to $NO_2$ from local shipping as relative contribution to all microenvironments (d-f), ME_home (g-i), ME_work (j-l) and ME_port (m-o) of absolute contribution from shipping related $NO_2$ exposure (a-c).**

## 4 Discussion of the generic exposure approach

We developed a generic approach to model population activity for exposure calculations (Sect. 2.6.2) to bridge the gap between static residency population numbers and very dynamic but specific population activity data derived from surveys or gathered with mobile devices, which were both not available in the harbour cities of this study. Thus, we used generic data and a set of assumptions, which introduces spatial and temporal uncertainties in the exposure calculation, additional to those of the applied CTM system. Exposure is the cross-product concentrations and population density. Therefore, all uncertainties that play a role for either of them have to be considered.

In terms of uncertainties within the applied CTM system to produce concentrations, the range of uncertainty can be identified by comparisons with measurements. The evaluation of measurements (Supplement SII, Table SII-2) shows a range of -26% to +4% for BIAS in annual measured vs. modelled $NO_2$ concentrations at different stations in Gdansk-Gdynia. In Rostock, there are higher underestimations of -56% to -32%, while in Riga the range is -60% to -4%. High underestimations in all cities mainly occur at or near traffic stations. Matthias et al. (2018) and Bieser et al. (2020) have shown, that the biggest uncertainty in CTM simulations are mostly due to emission data, which are a key driver and a major source of uncertainty to atmospheric chemistry transport models. Especially in urban areas, e.g. concentrations of $NO_x$ depend linearly on the local emissions. In emission modelling the amount, temporal and spatial distribution of emissions are often uncertain and thus have a high sensitivity. For example, NMVOC emissions for ships in port areas were not available as output from STEAM. This restriction led us to estimate NMVOC emissions based on the Carbon Monoxide (CO) emissions provided. Products of incomplete combustion, like CO and NMVOC, are difficult to estimate, because these emissions are very sensitive to engine load changes, engine control (mechanics/electronics), service history and fuel injection. Very little experimental information is available concerning NMVOC emissions from modern marine engines at sufficient level of detail and NMVOC emission factors based on measurements done decades ago may not represent NMVOC emissions from modern marine diesel engines accurately. Lack of detailed measurement data is probably because emission measurement standards (ISO 8178) do not require NMVOC classification, but report NMVOCs as total hydrocarbons instead, which makes evaluation of NMVOC species very difficult, hindering the CTM description of secondary aerosol formation at consecutive modelling effort. Nevertheless, in this study we used a CO emission to NMVOC emission ratio of 1.4, which is representative for emissions from auxiliary and main engines at an engine load of 70−80% (Aulinger et al., 2016), to calculate NMVOC emissions from STEAM CO emissions in Rostock, Riga and Gdansk-Gdynia. These uncertainties in emissions will translate to uncertainties in $NO_X$ concentrations due to the chemistry of ozone, $NO_X$ and volatile organic compounds (VOC), which represent one of the major uncertainties in the field of atmospheric chemistry, especially in urban areas (Sillman, 1999). Another example for uncertainties due to emissions are traffic emissions, which play a major role in the overall urban emissions. The exposures in the ME_traffic are very likely to be under-predicted in Rostock and probably also in Gdansk-Gdynia and Riga, due to the following reasons. In Rostock, the traffic emission modelling is not based on actual traffic density data but only was spatially disaggregated based on road type classification and corresponding factors, which represent a national average. While in Riga and Gdansk-Gdynia the traffic

emissions are based on traffic counts, they also do not account for all the effects of traffic congestion, slowing down of traffic in certain locations and streets and the effects of idling, and the deceleration and acceleration of vehicles. Traffic congestions can increase emissions in streets during rush hours (Gately et al., 2017; Requia et al., 2018; Smit et al., 2008). The evaluation at traffic stations has also shown that $NO_2$ was modelled with a high negative BIAS although EPISODE-CityChem was run with activated Street-Canyon-Module and therefore included treatment for dispersion in street canyons. The ME_port shows in all urban domains lower exposure to $NO_2$ compared to ME_work. This is mainly due to the detailed allocation of people directly employed by the port to the ME_port, which are distributed to the comparably large port areas.

Besides emissions, also meteorological fields and regional boundary conditions are crucial inputs for correct CTM simulations. Nevertheless, Karl et al. (2019a) have proven good agreement with measurements for the regional boundary conditions as calculated with CMAQ, and the performance of the meteorological module of TAPM shows very good agreements with measurements. Therefore having correct emissions is the highest priority in terms of improving the concentrations of $NO_2$, which then will linearly improve the results of exposure calculations.

In terms of uncertainties within the population activity, which is the second part of the cross product to calculate population exposure, there are four majors factors in the developed dynamic population activity approach that needs to be considered: the number of population, the temporal distribution of population, the spatial distribution of population and the application of infiltration factors for different microenvironments. In the following, these will be discussed in detail.

In this study, the population in each urban domain was derived from a population density map, valid for the European Union, instead of national or municipal population counts. This introduces biases in terms of total population numbers and the spatial distribution of people in their home environments. We have shown that the total population number derived from population density maps in this study is altered by 9%, 12% and 8% for Rostock, Riga and Gdansk-Gdynia respectively compared to population counts valid for the cities of interest (Table 3). Nevertheless, the advantage of this approach is the detachment from municipal boundaries or statistical zones, which are often used in population counts; these could lead to blind spots in research domains, which exceed municipal boundaries or statistical zones. A future development will be the integration of 'Population estimates by Urban Atlas polygon', which is a Copernicus Land Monitoring Service product in preparation (https://land.copernicus.eu/local/urban-atlas/population-estimates-by-urban-atlas-polygon, 06.02.2019). Besides this, we are uniformly distributing the derived total population with UA2012 land use classifications to spatially disaggregate the total population. A future development of this approach will be the integration of population density maps as a proxy in the distribution of population to the home environment, to integrate a weighted distribution of population to the UA2012 land use classifications. This will also lead to a clearer distinction of areas, which are allocated to work and home environments at the same time.

We considered the UA2012 land use classification "Continuous Urban Fabric" as both home and work environment with 30% and 70% share, due to the description of the UA2012 classification, which includes central business districts. To check the impact of this assumption, we changed the applied split of 30% ME_home and 70% ME_work, in two tests to (1) 50% ME_home and 50% ME_work and to (2) 70% ME_home and 30% ME_work in the Gdansk-Gdynia domain. By changing

the distribution of ME_home to 50%, the contribution of ME_home to the total annual gridded mean increases by 0.7%, while the total annual exposure increases by 1.8%. Changing the distribution of ME_home to 70%, increases the contribution of ME_home to the total annual gridded mean by 1.2%, while the total annual exposure increases by 3.2%. In the same tests, the ME_work is changed to 50% and 30%, which results in a decrease of the ME_work contribution to the annual grid mean by 0.3% and 0.5%. Therefore, we evaluate the uncertainty of the applied split of 70% ME_work and 30% ME_home in the UA2012 land use class "Continuous Urban Fabric" to have limited influence on the overall exposure results. Nevertheless, due to a lack of information about specific population activity in any of the urban domains, we cannot validate our assumptions in distributing population to the MEs and the connected UA2012 land use classifications. Based on the descriptions of the UA2012 land use classifications we matched the best fitting microenvironments but still introduce uncertainties, e.g. in the category "Industrial, commercial, public, military and private units" which contains not only work environments but also non-work environments, e.g. schools, universities, museums or churches. When it comes to ME_work, we also considered the UA2012 class "Continuous urban fabric" to mainly constitute indoor work environments in city centres and the UA2012 classes "Industrial, commercial, public, military and private units", "Mineral extraction and dump sites" and "Construction Sites" to account for mixed indoor and outdoor work environments. In future studies, a clearer distinction of the UA2012 categories in terms of numbers of workers and indoor/outdoor classification should be done; e.g. the number of workers in the category "Mineral extraction and dump sites" could be taken from city-specific statistics and the category could be classified as outdoor only environment. Besides this, we considered the amount of commuters, taken from municipal statistics, in the ME_work and ME_traffic and thus accounted for people, which are additionally exposed to pollution in traffic and work environments. The consideration of commuters in Gdansk-Gdynia leads to a 4% higher total annual population exposure and a 20% higher annual exposure in ME_work. For a better distribution of the ME_work and ME_other we plan to use the "point of interest" feature in OSM data as proxy in future studies, which potentially allows for a better distribution between work and other activities and to identify very busy city-centres.

Besides uncertainties in the spatial distribution, we also introduced uncertainties regarding the temporal distribution, which is based on a temporal profile for the city of Helsinki (Soares et al., 2014). We adapted this profile and then added features, which we found to appear in other European cities, such as traffic rush hours in the morning and evening. However, such a generic profile is not able to reflect the actual population activity throughout the day. Moreover, there are regional and national differences, e.g. the siesta in Mediterranean countries. Still this pattern emulates a dynamic population, which moves between environments and is exposed to different levels of pollution throughout the day. In comparison to traditional approaches, which assume people to be at their residence (home address) all the time, we believe this approach is beneficial in particular for cities in European regions where data from surveys or positioning data from mobile devices is missing. We compared population exposure to $NO_2$ based on our dynamic population activity approach, with population exposure based on a static approach to analyse the effect of a population moving in space and time on calculated population exposures. In this test, we allocated the total population all day (100% of the time) to the home environment (ME_home) in order to simulate a static approach. The dynamic activity considers people 'moving' diurnally between different MEs. Moreover, we ran

simulations with and without infiltration factors to test the effect of outdoor concentrations infiltrating to indoor environments in the static and dynamic approach. The comparison between the static and the dynamic approach without the consideration of IF (i.e. indoor air concentrations are the same as in the surrounding outdoor air) shows a decrease in total annual exposure in each city (Table 6). Therefore, the consideration of diurnal dynamic activity in different MEs leads to an increase in total population exposure. This is an effect of people moving to areas, which are more polluted, and additionally the effect of commuting inside/outside of the city.

Another assumption made in calculating exposure in different environments is the infiltration of outdoor pollutant concentrations into indoor environments. We have considered the influence of outdoor air pollution on the total population exposure. However, we have not addressed indoor sources and sinks of pollution although, indoor sources such as, e.g. tobacco smoking, cooking, heating and cleaning might cause additional short-term concentration maxima in indoor environments. We have also assumed that infiltration is temporally constant, changing only with the seasons. Nevertheless, we took into account the infiltration of outdoor pollution into indoor environments (ME_work and ME_home) using IFs. To check the impact of IFs for the indoor environments, we increased and lowered the applied IFs in ME_work and ME_home in the city of Gdansk. An increase of the IFs by 0.1 in both MEs leads to a linear increase of 10% in ME_home and ME_work respectively. The total exposure increases by 10%. When it comes to the relative contribution of each ME to the total exposure, the relevance of ME_home increases to 57% (+2.5% points) and ME_work to 14% (+0.4% points). A likewise decrease of IFs by 0.1 shows the same changes with opposite sign. Thus, the impact of the adapted IFs on exposure in environments that are mostly indoor environments has a significant influence on the total exposure results with a linear response of the total exposure to changes of the IF. The MEs ME_other, ME_traffic and ME_port are considered outdoor environments. When it comes to the ME_other, which is an outdoor-only environment in this study, the exposure is heavily dependent on the season, due to more people spending their time outdoors in summer than in winter. This has not been considered in this study but should be taken into account in future studies. Nevertheless, the ME_other areas in the city-centre are mainly green urban areas and therefore in summer potentially areas of high exposure. In general, the applied IFs for $NO_x$ as derived from Borrego et al. (2009) are representing an average of infiltration measurements in Korea (Baek et al., 1997), Hongkong (Chau et al., 2002) and the United Kingdom (Dimitroulopoulou et al., 2006). Thus, in future studies it is desirable to derive and use IF, which are representative for the city-specific building infrastructure to account for different air-intake techniques, building structures or different ventilation manners. Better parametrization to derive more representative IF could be derived from a combination of the EU Buildings Database, the UA2012 and climate data.

Taking into account all uncertainties and possibilities for improvement, we promote this approach for European regions, in which actual data on population activity is not available, with the overall goal to improve existing exposure calculations for policy support. Nevertheless, the highest uncertainties and therefore possibilities to improve the results of the exposure calculations are

1. a better representation of emission inventories in CTM,

2. city- and microenvironment-specific infiltration factors for indoor environments,

3. city- and microenvironment-specific time profiles of population activity, and

4. city-specific spatial distribution of population in representative microenvironments.

## 5 Conclusions

We have presented population exposure to total and shipping related $NO_2$ outdoor concentrations in different microenvironments of the Baltic Sea harbour cities Rostock, Riga and the urban agglomeration of Gdansk-Gdynia. The population exposure was calculated as a product of (1) hourly-varying surface concentrations of $NO_2$ simulated with a global-to-local chemistry transport model chain and (2) a newly developed generic approach to account for dynamic population activity in European cities.

We simulated the surface concentrations with the urban-scale CTM EPISODE-CityChem using regional boundary conditions from CMAQ simulations, land-based and ship emissions and meteorological fields for 2012 in Rostock, Riga and Gdansk-Gdynia. The evaluation of modelled versus measured $NO_2$ time series showed good spatial correlations, slight underestimations of annual NO2 but an overall applicable performance for studies in urban areas with a FAC2 value above 0.3 at all stations of each domain. The simulated results for show contribution of $NO_2$ from shipping to overall air quality 22% for Rostock, 11% for Riga and 16% for Gdansk-Gdynia.

We developed a generic dynamic approach to account for population activity in European urban areas, which is applicable for exposure calculations. Our approach aims at filling the gap between traditional approaches of exposure calculations, which are based on static population counts at residential addresses, and approaches, that take into account individual activities as derived from surveys or individual GPS data. Due to missing surveys and individual GPS data in the research domains of this study, we combined existing, publicly available data, to follow state-of-the art exposure modelling approaches in four steps. At first, we split the total population of each urban domain into several microenvironments (home, work, traffic, other, port). Second, we distributed these microenvironments to matching land use classifications of the Urban Atlas 2012. Third, we temporally distributed the total population to the different microenvironments diurnally for weekdays and weekends, adapted from existing diurnal patterns in other European cities. Fourth, we applied infiltration factors for indoor environments, to account for outdoor concentrations infiltrating indoor environments. Following this approach, it is possible to compile gridded datasets containing the number and spatial distributions of a city's population for every hour in a diurnal cycle in each defined microenvironment. For this study, we generated these grids with a grid resolution of 100 m, following the resolution of the simulated surface concentration.

In the exposure calculation, we focused on exposure to $NO_2$, because the ship influence was shown to be high and the regulations for $NO_x$ emission reductions will propagate slowly into the ship fleet. Moreover, $NO_2$ from ships adds to other local sources and therefore brings problems to obey AQ Directive targets of annual mean $NO_2$. Besides this, outdoor $NO_2$ pollution is a health concern with lot of recent attention by the WHO.

The relative contribution of each microenvironment to total $NO_2$ exposure is highest for the home environment with 59%, 54% and 55% in Rostock, Riga and Gdansk-Gdynia respectively. Although the home environment has shown to be very sensitive to applied infiltration factors, the vast amount of people spending their time at home during the day makes the home environment the most important environment in terms of exposure to outdoor $NO_2$. When it comes to the influence of local shipping activities, shipping contributes with 13%, 6% and 4% to $NO_2$ exposure in all microenvironments in Rostock, Riga and Gdansk-Gdynia. The shipping contribution mainly focuses on MEs near the port in all cities. MEs, which are close to the port areas, can be influenced by shipping with up to 80% in Rostock and Riga and up to 50% in Gdansk-Gdynia. The lower contributions in Gdansk-Gdynia are due to $NO_2$ concentrations from shipping transported towards the open sea with the predominating south westerly winds, while in Rostock and Riga the home and work environments north of the port are mainly affected from shipping for the same reason. The differences in relative contributions from shipping are determined by the magnitude of shipping activities in relation to activities in the rest of the domain and the domain size. The contribution of shipping in the port environment is considerably higher with 46%, 44% and 26% respectively. Nevertheless, the port environment stands for less than 1% of the total exposure in all domains.

In general, the applied approach for exposure modelling is capable of showing the diurnal variation of population activity and therefore diurnal exposure in different microenvironments although we focused on total annual population exposure in this study. By introducing dynamic population activity instead of static population activity, the total exposure in Rostock, Riga and Gdansk increases and therefore illustrates the need to consider dynamic population activity in exposure studies. In addition, we demonstrated the importance of microenvironment- and region-specific infiltration factors to consider outdoor concentrations infiltrating indoor environments. The lack of city-specific activity profiles, workplace addresses and infiltration factors introduces the biggest uncertainties in this study. In future studies we plan to improve the spatial allocation of population by applying population density maps in the spatial disaggregation of people in the home environment and by applying OSM points of interest as well as sector statistics on workers. Thereby, a better differentiation of infiltration factors in the work environments appears to be feasible. Moreover, we plan to integrate parametrizations for infiltration factors, which will take into account public national data on building structures and building regulations as well as climate data. When it comes to the traffic environment, we also aim at integrating region-specific measurements of outdoor to indoor concentration ratios. Besides these efforts, further studies to test the impact of different emission sectors, such as traffic or industry, in different microenvironments are planned.

The developed and first-time applied approach for generic dynamic population activity for calculating exposure to surface concentrations advances over traditional static approaches and can be transferred to other cities in Europe since no need for local activity profiles is involved. Although we used a global-to-local chemistry transport model chain, the presented generic dynamic population calculation can also be used with surface concentrations field created with other methods. Therefore, we promote this approach for European regions, in which specific population activity data derived from surveys or gathered with mobile devices is not available, with the overall goal to improve existing exposure calculations for policy support and to provide the basis for health effect studies.

**Data availability**

The following data sets are available for download from the HZG ftp server upon request: (1) input data for the one-year AQ simulations of Rostock, Riga and Gdansk-Gdynia (full set ca. 100 GB); (2) DELTA Tool data for comparison of model output and measurements; (3) model output data of the AQ simulations of Rostock, Riga and Gdansk-Gdynia (full set ca. 100 GB); (4) model input and output data of the exposure calculations for all microenvironments of Rostock, Riga and Gdansk-Gdynia (full set ca. 100 GB).

**Appendix A: Statistical indicators and model performance indicators**

In the statistical analysis of the model performance, the following statistical indicators are used: normalized mean bias (NMB), standard deviation (STD), root mean square error (RMSE), correlation coefficient (Corr), index of agreement (IOA) and the fraction of predictions within a factor of two of observations (FAC2). The overall bias captures the average deviations between the model and observed data and the normalized mean bias is given by:

(A1) $\qquad NMB = \frac{\overline{M}-\overline{O}}{\overline{O}}$ ,

where $M$ and $O$ stand for the model and observation results, respectively. The overbars indicate the time average over $N$ time intervals (number of observations). The root mean square error combines the magnitudes of the errors in predictions for various times into a single measure and is defined as:

(A2) $\qquad RMSE = \sqrt{\frac{1}{N} * \sum_{i=1}^{N}(M_i - O_i)^2}$ ,

where subscript i indicates the time step (time of observation values). RMSE is a measure of accuracy, to compare prediction errors of different models for a particular data and not between datasets, as it is scale-dependent. The correlation coefficient (Pearson r) for the temporal correlation is defined as:

(A3) $\qquad r = \frac{\sum_{i=1}^{n}(O_i-\overline{O})\cdot(M_i-\overline{M})}{\sqrt{\sum_{i=1}^{n}(O_i-\overline{O})^2 \cdot \sum_{i=1}^{n}(M-\overline{M})^2}}$ ,

including the standard deviation of model STDM and observation STDO data, respectively. The standard deviations are:

(A4) $\qquad STDM = \sqrt{\frac{1}{N-1} * \sum_{i=1}^{N}\left(M_i - \overline{M}\right)^2}$

(A5) $\qquad STDO = \sqrt{\frac{1}{N-1} * \sum_{i=1}^{N}\left(O_i - \overline{O}\right)^2}$

The index of agreement is defined as:

(A6) $$IOA = 1 - \frac{\sum_{i=1}^{N}(O_i - M_i)^2}{\sum_{i=1}^{N}(|M_i - \bar{M}| + |O_i - \bar{O}|)^2}$$

An IOA value close to 1 indicates agreement between modelled and observed data. The denominator in Eq. (A6) is referred to as the potential error. The fraction of modelled values within a factor of two (FAC2) of the observed values are the fraction of model predictions that satisfy is defined as:

(A7) $$0.5 \leq \frac{M_i}{O_i} \leq 2.0$$

For evaluation of modelled values in rural areas, the acceptance criteria is FAC2 ≥ 0.5, while in urban areas it is FAC2 ≥ 0.3 (Hanna & Chang 2012). The indicator $H_{perc}$ for the model capability to reproduce extreme events, e.g. exceedances is defined as:

(A8) $$H_{perc} = \frac{|M_{perc} - O_{perc}|}{\beta U_{95}(O_{perc})} \quad \text{and} \quad MPC: H_{perc} \leq 1$$

Where *"perc"* is the selected (high) percentile, $M_{perc}$ and $O_{perc}$ are the modelled and observed values corresponding to the selected percentile (Thunis et al. 2012).

**Author Contribution**

M.O.P. R. created the overall structure, prepared meteorological and emission input data for the EPISODE-CityChem simulations, performed and evaluated the EPISODE-CityChem concentration simulations, developed and applied the generic dynamic activity approach, visualised and plotted all results, and wrote major parts of this publication. M. K. assisted with writing and discussing the overall structure, did the setup of the EPISODE-CityChem for all domains and programmed the pre-processing utilities. J. B. created land-based emissions with the SMOKE-EU model and contributed text on land-based emissions in chapter 2.4. J.-P. J. & L. J. created local shipping emissions with the STEAM model and contributed text on shipping emissions in chapter 2.5.

**Competing interests**

The authors declare that they have no conflict of interest.

**Acknowledgements**

This work is part of the BONUS SHEBA (Sustainable Shipping and Environment of the Baltic Sea region) research project under Call 2014-41. BONUS (Art 185) is funded jointly by the EU, Innovation Fund Denmark, Estonian Research Council, Academy of Finland,  and by the German Federal Ministry of Education and Research under Grant Number 03F0720A, National Centre of Research and Development (Poland) and Swedish Environmental Protection Agency.

We acknowledge Michalina Bielawska (ARMAAG), Iveta Steinberga (ELLE, University of Latvia), Stefan Nordmann and Stefan Feigenspan (UBA) for the preparation and distribution of emission datasets for Gdansk-Gdynia, Riga and Rostock. Christane Gackenholz (former HZG) is thanked for the preparation of emission data for the UECT pre-processing utilities. Moreover, we would like to thank Stefan Seum (DLR) for traffic data from the VEU project. Copernicus Services is thanked for the public distribution of Urban Atlas and population density products. Open Street Map is thanked for maps used in plots and open source road data, which was used to distribute traffic emissions. The air quality model CMAQ is developed and maintained by the U.S. Environmental Protection Agency (US EPA). COSMO-CLM is the community model of the German climate research. The simulations with COSMO-CLM, CMAQ, EPISODE-CityChem and the exposure calculations were performed at the German Climate Computing Centre (DKRZ) within the project "Regional Atmospheric Modelling" (Project Id 0302).

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

**Table 1: Overview of EPISODE-CityChem setup, the TAPM meteorological setup and emission data for each urban domain.**

| | Gdansk-Gdynia | Riga | Rostock |
|---|---|---|---|
| **CTM setup with EPISODE-CityChem** | | | |
| CTM domain extent | $40 \times 40$ km² | $20 \times 20$ km² | $16 \times 16$ km² |
| CTM grid resolution | 1000 m | 400 m | 400 m |
| Boundary Conditions | Interpolated from regional CMAQ simulation in the North and Baltic Sea 2012 with 4 km x 4 km spatial and 1-hour temporal resolution (Karl et al. 2018). | | |
| **Meteorological setup with TAPM** | | | |
| Synoptic scale data four outer domain forcing | Three-hourly synoptic scale ECMWF ERA5 reanalysis ensemble means on a longitude/latitude grid at 0.3 degree grid spacing. | | |
| Meteorological domain extent | $40 \times 40$ km² | $20 \times 20$ km² | $16 \times 16$ km² |
| Meteorological grid resolution | 1000 m | 400 m | 400 m |
| Land cover database | CLC 2012 | CLC 2012 | CLC 2012 |
| Terrain height database | EU-DEM | EU-DEM | EU-DEM |
| **Boundary conditions** | CMAQ simulation with 4 km grid resolution on hourly basis. | | |
| **Emission inventories** | | | |
| Shipping | Hourly emissions with grid resolution of 250 m, in two height layers (0<36 m, >=36 m<1000 m) from STEAM | | |
| Point (energy and combustion) | 676 sources (ARMAAG) | 2,875 sources (ELLE) | 32 sources (LUNG) |
| Area (residential heating, agriculture, solven use) | Interpolation of 4 km resolution SMOKE-EU | Interpolation of 4 km resolution SMOKE-EU | 400 m resolution UBA emission inventory |
| Line (traffic) | 9,884 sources (ARMAAG) | 2,875 sources (ELLE) | 3,875 sources (UBA, OSM, VEU) |

**Table 2: Mapping of Urban Atlas 2012 classification with selected microenvironments and infiltration factors (IF) for indoor-outdoor relationships in winter (Sep-Feb) and summer (Mar-Aug) months.**

| Code | UA2012 classification | Microenvironment | IF Winter $NO_x$ | IF Summer $NO_x$ |
|------|----------------------|------------------|------------------|------------------|
| 11100 | Continuous Urban Fabric | 30% ME_home | 0.7 [a] | 0.8 [a] |
| | | 70% ME_work | 0.75 [a] | 0.85 [a] |
| 11210 | Discontinuous Dense Urban Fabric | ME_home | 0.7 [a] | 0.8 [a] |
| 11220 | Discontinuous Medium Density Urban Fabric | ME_home | 0.7 [a] | 0.8 [a] |
| 11230 | Discontinuous Low Density Urban Fabric | ME_home | 0.7 [a] | 0.8 [a] |
| 11240 | Discontinuous Very Low Density Urban Fabric | ME_home | 0.7 [a] | 0.8 [a] |
| 11300 | Isolated Structures | ME_home | 0.7 [a] | 0.8 [a] |
| 12100 | Industrial, commercial, public, military, private units | ME_work | 0.75 [a] | 0.85 [a] |
| 13100 | Mineral extraction and dump sites | ME_work | 0.75 [a] | 0.85 [a] |
| 13300 | Construction Sites | ME_work | 0.75 [a] | 0.85 [a] |
| 12300 | Port areas | ME_port | 1 [b] | 1 [b] |
| 12210 | Fast transit roads and associated land | ME_traffic | 1 [b] | 1 [b] |
| 12220 | Other roads and associated land | ME_traffic | 1 [b] | 1 [b] |
| 14100 | Green urban areas | ME_other | 1 [b] | 1 [b] |
| 14200 | Sports and leisure facilities | ME_other | 1 [b] | 1 [b] |

[a] (Borrego et al., 2009; Baek et al., 1997; Chau et al., 2002; Dimitroulopoulou et al., 2006), [b] estimate in this study

**Table 3: Statistical Data for 2012 to refine population distribution in the research domains.**

| | Population [1000 habitants] | | Employment rate | Commuter [habitants] | Port Work [# workers] | Port Turnover [Mio. t] |
|---|---|---|---|---|---|---|
| | City Statistics | CLC [d] | | | | |
| Rostock | 203 [a] | 222 (+9%) | 52% [a] | 10 k [a] | 2600 [ej] | 21,2 [f] |
| Riga | 699 [b] | 784 (+12%) | 66% [b] | 90 k [b] | 6000 [ej] | 36,1 [g] |
| Gdansk | 796 [c] | 861 (+8%) | 51% [c] | 32 k [c] | 3300 [e] | 26,9 [h] |
| Gdynia | | | | 19 k [c] | 2600 [e] | 15,8 [i] |
| Helsinki | 600 [d] | - | 62% | | - | - |

Sources: [a] Hanse- und Universitätsstadt Rostock, [b] Riga City Council City Development Department, [c] Statistics Poland, [d] Population density disaggregated with Corine land cover 2000 (Gallego, 2010), [e] European Commission Maritime Affairs, [f] Rostock Port, [g] Freeport of Riga, [h] Port Gdansk, [i] Port Gdynia, [j] own calculation

5  **Table 4: Summary of shipping impact on $NO_2$ and $PM_{2.5}$ concentrations as total annual averaged grid mean for the total domains in 2012.**

| Rel. Ship Influence | NO$_2$ | PM$_{2.5}$ |
|---|---|---|
| **Rostock** | 22% | 1% |
| **Riga** | 11% | 1% |
| **Gdansk-Gdynia** | 16% | 3% |

**Table 5: Total annual averaged grid mean exposure to NO$_2$ due to shipping emissions in different microenvironments as relative to the total annual averaged grid mean exposure to NO$_2$ from all sources.**

| Rel. Ship Influence NO$_2$ | Rostock | Riga | Gdansk-Gdynia |
|---|---|---|---|
| **All Microenvironments** | 12.7% | 5.5% | 4.4% |
| **Home** | 13.8% | 5.5% | 3.6% |
| **Work** | 9.9% | 5.2% | 4.3% |
| **Port** | 45.6% | 43.9% | 26.4% |
| **Traffic** | 10.6% | 4.4% | 3.4% |
| **Other** | 10.7% | 5.9% | 6.0% |

**Table 6: Comparison of total exposure to NO$_2$ in each city for simulations with static and dynamic population, with and without ME- and seasonal specific IF. The approach used in this study (Dynamic activity with IF) is representing the baseline (100%).**

| Scenario | Rostock | | Riga | | Gdansk-Gdynia | |
|---|---|---|---|---|---|---|
| | Total NO$_2$ exposure [µg m$^{-3}$ * pop] | Rel. change to baseline | Total NO$_2$ exposure [µg m$^{-3}$ * pop] | Rel. change to baseline | Total NO$_2$ exposure [µg m$^{-3}$ * pop] | Rel. change to baseline |
| Dynamic Activity with IF | 9.15 E+09 | (baseline) | 6.55 E+10 | (baseline) | 7.66 E+10 | (baseline) |
| Dynamic Activity without IF | 1.25 E+10 | + 27% | 8.88 E+10 | + 26% | 9.85 E+10 | +22% |
| Static Activity with IF | 8.89 E+09 | - 3% | 6.02 E+10 | - 9% | 6.88 E+10 | -11% |
| Static Activity without IF | 1.19 E+10 | + 23% | 8.03 E+10 | + 18% | 9.18 E+10 | +17% |