# Peer review of "Urban population exposure to $NO_X$ emissions from local shipping in three Baltic Sea harbour cities – a generic approach"

_Atmospheric Chemistry and Physics, 2019_

## Referee Comment (RC1) · Fabian Lenartz (Referee) · 17 Apr 2019

Dear Authors,

Congratulations for your work. It was an interesting reading and I believe that the generic approach you proposed to account for dynamic population will be used again in future exposure studies. The thorough description of the methodology is helpful to understand the results and support the conclusions. The content of the paper is somewhat richer than the title suggests, I would recommend to modify the title or the text accordingly. For example, if one focuses only on NOx emissions are the second part of Section 2.5 (pg 10 l.25 to pg 11 l.3) and the last part of Section 3.2 (pg 18 l.23 to

28) relevant? Furthermore, I feel like one studies an exposure to concentrations rather than to emissions, but I know that such expression can be found in the litterature. The overall presentation is well structured, the language seems fluent and precise for the non-native speaker who I am and the quality of the figures is good.

Here below you will find two questions about your work: - If you wanted to reduce uncertainty on the presented results, how would you prioritize the following tasks improvement of emission inventories, improvement of model performance (better fit with observed values, higher spatial resolution, etc.), use of more precise/diverse activity patterns, use of more precise/diverse infiltration factors, etc.? - Have you tested the impact on exposure of a different emission sector such as traffic?

Finally, here are some minor comments about the article: - pg 1 l.15 exposure TO outdoor (...) - pg 7 l.26 grid resolution of 4 km or 2 km but not 4 km2 - pg 10 l.21 I think the first reference should be Hulskotte and Denier van der Gon, 2010 - pg 10 l.28 mechanics/electronics instead of mechanic/electronic - pg 10 l.30 modeling is here used with one l instead of two - pg 16 l.1 I'd stop the sentence after "the observed values" (+ FAC2) - pg 27 l.16 I wouldn't include a reference in the conclusions - pg 27 l.20 to 22 I wouldn't keep the part about PM10 and PM2.5 - pg 27 l.29 a four-step approach is mentioned in the conclusions while a five-step one is mentioned in Section 2.6.2 - pg 29 l.11 the code for exposure modelling should be made available before publication or this sentence should be deleted

---

## Referee Comment (RC2) · Anonymous Referee #2 · 1 May 2019

This study was an example investigating the urban population exposure to local shipping emissions. To raise a generic approach, the authors started from the very beginning including the built up pf emission inventory and spatial-temporal allocations for high resolution modeling. To obtain the health impacts, exposure responses were also studied. As shown in the title, NOx is the main target although other pollutants were also discussed. The study is comprehensive and flawless from the structure to presentation. Overall, this manuscript is well organized. This topic is relevant to the scope of ACP and also in time addressing the pollutant from local shipping emissions. Thus I recommend publication of this paper within ACP. As mentioned by the other reviewer, this manuscript is a little bit longer than the regular ones. Shall the authors consider to

put some materials in the supplemental materials or refer to some other previous studies? For example, the built up of emission inventory or the spatial-temporal allocation? If possible, comparisons with similar EI studies would be very helpful. Uncertainties of the NMVOC and the impacts on NOx simulation should also be discussed. The uncertainties come from the different steps should be discussed too. For example, the EI, air quality model, exposure responses etc? Any seasonal or monthly differences for your results? Or during the shipping busy/non-busy periods, how about the sensitivity of the simulations? Minors: Page 1 Line 18, page 12 line 9: 100X100 functions should be used instead of letter x Page 18: NO2 should use subscript.
* * *

---

## Author Comment (AC2) · 31 May 2019

We attached the author's response as supplement. Thank you for your review and best regards.

Please also note the supplement to this comment:
https://www.atmos-chem-phys-discuss.net/acp-2019-127/acp-2019-127-AC2-supplement.pdf

---

## Author Response (AR1)

| | |
|---|---|
| **Red Shading:** | **Comments from referees** |
| **Orange Shading:** | **Author's response** |
| **Purple Shading:** | **Author's changes in manuscript** |

**Authors response to Report #1 – Fabian Lennartz (R1)**

**We are very grateful to Fabian Lennartz for his positive review of the manuscript. Below his remarks, comments and question are addressed in detail. We are in particular grateful for questions addressing the uncertainty of the method, which led to changes in the manuscript and helped to improve the manuscript.**

*R1.0. "Congratulations for your work. It was an interesting reading and I believe that the generic approach you proposed to account for dynamic population will be used again in future exposure studies. The thorough description of the methodology is helpful to understand the results and support the conclusions. [...] The overall presentation is well structured, the language seems fluent and precise for the non-native speaker who I am and the quality of the figures is good."*

**Response to R1.0:**

**We thank the reviewer for his assessment of the scope, methodology and structure of the manuscript.**

*R1.1. "The content of the paper is somewhat richer than the title suggests, I would recommend to modify the title or the text accordingly. For example, if one focuses only on NOx emissions are the second part of Section 2.5 (pg 10 l.25 to pg 11 l.3) and the last part of Section 3.2 (pg 18 l.23 to relevant?"*

**Response to R1.1:**

**We aware of the extent of the paper and decided to move some parts to the annex, to focus on the scope suggested by the title and the reviewers suggestions:**

**We narrowed down the chapter on evaluation of concentrations in the manuscript and moved the detailed evaluation into the Annex.**

**This led to a change in chapter 3.1 in the manuscript in form of narrowing it to:**

"Due to an insufficient number of valid time series at the measurement stations in 2012 for Rostock and Riga to achieve significant performance indication, we focus on a discussion of measurement evaluation in the Gdansk-Gdynia agglomeration, which contains eight valid $NO_2$ measurement time series. In Rostock, there are four stations for $NO_2$, while in Riga there are two stations for $NO_2$. However, statistical indicators for $NO_2$, $O_3$ and $PM_{10}$ for all available stations in all cities as well as a detailed description of the AQ simulation performance in Rostock and Riga can be found in Supplement II of this paper.

The analysis of spatial correlations for $NO_2$ time series in Gdansk-Gdynia has shown an r² of 0.3 for station averaged daily averages in 2012 and an r² of 0.79 for station-specific annual averages (Figure 8). The analysis of temporal correlation for hourly values over one year at single stations shows four urban background stations with r values between 0.3 and 0.35 and four urban background stations with r values between 0.2 and 0.3. The poorer correlation values can be expected due to not-localised information on temporal emissions. Modelled $NO_2$ for hourly values over one year is in agreement with observed $NO_2$ with overestimation of $NO_2$ at station Wrzescez (urban background station located in an urban green area, Latitude 54.38028, Longitude 18.62028, height asl 40 m) by 4% and underestimation of $NO_2$ (-1% to – 26%) at all other (urban

background) stations. NO$_2$ shows overall good performance and FAC2 values for NO$_2$ in Gdansk-Gdynia reach from 0.46-0.7 and are therefore fulfilling the acceptance criteria for urban regions of FAC2 $\geq$ 0.3 as defined by Hanna and Chang (2012).

[Figure]

**Figure 1: Modelled versus measured NO$_2$ concentrations at all available measurement stations in the Gdansk-Gdynia research domain. (a) Shows annual station averages with each dot indicating one station, while (b) shows daily averages with each dot indicating an average of all stations. For (a) and (b) the colours display seasons."**

**The part about influence of shipping by PM$_{2.5}$ (pg 18 l.23), is deleted from the manuscript.**

**When it comes to the second part of section 2.5 (pg 10 l.25 to pg 11 l.3), which deals with uncertainties in the NMVOC emission inventory for shipping, we decided to move this section to the discussion part of the manuscript, to discuss uncertainties and improvement possibilities. This is necessary due the chemistry of ozone, NO$_X$ and volatile organic compounds (VOC), which represents one of the major uncertainties in the field of atmospheric chemistry, especially in urban areas (Sillman, 1999). The discussion of uncertainties due to NMVOC emissions and chemistry was also requested by Reviewer 2 (R2.3) and tackles the R1.3.**

**Therefore, the respective sentences in section 2.5 have been moved from to the updated discussion section (seer response R1.3).**

**Besides these reductions of the manuscript's extent, we believe that the amount of given information is necessary to follow:**

    **a.** **the preparation of emissions and concentrations as well as their evaluation with common methods,**

    **b.** **the development of the generic approach for population activity exposure assessments and**

    **c.** **the discussion of uncertainties in population exposure, which is connected with modelled concentrations and population activity (as requested in more detail by Reviewer 1 and 2).**

*R1.2. "Furthermore, I feel like one studies an exposure to concentrations rather than to emissions, but I know that such expression can be found in the literature."*

**Response to R1.2:**

**Indeed, we study the exposure to concentrations, which are a result from shipping emissions. Nevertheless, we decided not to change the title, due to the common use of this terminology in literature, and the close relationship of emissions and concentrations, especially on the urban scale.**

**Nevertheless, we clarified the study scope by changing the first sentence of objectives in the introduction to: "The objective of this study is to identify the impact of emissions due to local shipping activities on air quality and population exposure to concentrations of NOx in three major Baltic Sea harbour cities: Rostock (Germany), Riga (Latvia) and the urban agglomeration of Gdansk-Gdynia (Poland)."**

*R1.3. "Here below you will find two questions about your work: If you wanted to reduce uncertainty on the presented results, how would you prioritize the following tasks improvement of emission inventories, improvement of model performance (better fit with observed values, higher spatial resolution, etc.), use of more precise/diverse activity patterns, use of more precise/diverse infiltration factors, etc.?"*

**Response to R1.3:**

**In the following, we will discuss the review comments R1.3, R2.3 and R2.4 by reviewers 1 and 2 jointly, due to the similar nature of the comments. The comments deal with questions of uncertainty in the developed generic approach. We decided to dedicate a paragraph to this issue, integrated additional information and parts of the existing manuscript to restructure the discussions chapter 4 as follows to replace the existing chapter 4 in the reviewed manuscript:**

**Short answer in the order of priority:**

[revised manuscript text omitted]

*R1.4. "Have you tested the impact on exposure of a different emission sector such as traffic?"*

**Response to R1.4:**

**Not yet, but further studies with improvements of microenvironments and focus to different sectors, especially traffic, are planned. A study on the integration of better and region-specific infiltration factors in the traffic environment is in**

10 **progress.**

**Thus, we integrated the following sentences in the outlook section of the paper:**

**"When it comes to the traffic environment, we also aim at integrating region-specific measurements of outdoor to indoor concentration ratios. Besides these efforts, further studies to test the impact of different emission sectors, such as traffic or industry, in different microenvironments are planned."**

**List of Minor changes requested by R1 have been considered in the final manuscript:**

"pg 1 l.15 exposure TO outdoor (...)"

  o   **changed to "exposure to outdoor […]"**

"pg 7 l.26 grid resolution of 4 km or 2 km but not 4 km2"

20  o   **changed to "4 x 4 km²"**

"pg 10 l.21 I think the first reference should be Hulskotte and Denier van der Gon, 2010"

  o   **changed to "Hulskotte and Denier van der Gon, 2010"**

"pg 10 l.28 mechanics/electronics instead of mechanic/electronic"

  o   **changed to "mechanics/electronics"**

"pg 10 l.30 modeling is here used with one l instead of two"

- o **changed to "modeling"**

"pg 16 l.1 I'd stop the sentence after "the observed values" (+ FAC2)"

- o **the sentence now stops as suggested by R1**

"pg 27 l.16 I wouldn't include a reference in the conclusions"

- o **we excluded all references from the conclusions, which are unnecessary repetitions of previous mentioned references**

"pg 27 l.20 to 22 I wouldn't keep the part about PM10 and PM2.5"

- o **We deleted the part about PM10 and PM2.5 due to the scope of the paper, which is NOx. Therefore results on PM10 and PM2.5 are excluded from the conclusions.**

"pg 27 l.29 a four-step approach is mentioned in the conclusions while a five-step one is mentioned in Section 2.6.2"

- o **We changed the respective parts to "a four-step approach"**

- "pg 29 l.11 the code for exposure modelling should be made available before publication or this sentence should be deleted."

**While the code is still in preparation for publication, we deleted the sentence about code availability.**

**Author's response to Report # 2 – anonymous referee (R2)**

**We are very grateful to the anonymous reviewer of the manuscript. Below her/his remarks, comments and question are addressed. We are in particular grateful for questions addressing the uncertainty of the method, which led to changes in the manuscript and helped to improve the manuscript.**

*R2.0. "This study was an example investigating the urban population exposure to local shipping emissions. To raise a generic approach, the authors started from the very beginning including the built up pf emission inventory and spatial-temporal allocations for high resolution modeling. To obtain the health impacts, exposure responses were also studied. As shown in the title, NOx is the main target although other pollutants were also discussed. The study is comprehensive and flawless from the structure to presentation. Overall, this manuscript is well organized. This topic is relevant to the scope of ACP and also in time addressing the pollutant from local shipping emissions. Thus I recommend publication of this paper within ACP."*

**Response to R2.0:**

**We thank the reviewer for her/his assessment of the scope, methodology and structure of the manuscript.**

*R2.1. "As mentioned by the other reviewer, this manuscript is a little bit longer than the regular ones. Shall the authors consider to put some materials in the supplemental materials or refer to some other previous studies? For example, the built up of emission inventory or the spatial-temporal allocation?"*

**Response to R2.1:**

- **Please see response R1.1 to reviewer 1 for a similar request.**
- **We decided to keep the built-up of emissions inventory incl. the spatial-temporal allocation because they are a key driver and a major source of uncertainty to atmospheric chemistry transport models (Bieser et al., 2020, Matthias et al., 2018).**

10 *R2.2. "If possible, comparisons with similar EI studies would be very helpful."*

**Response to R2.2:**

**Enquiries beforehand this study resulted in only few existing studies on population exposure to NO$_X$ in urban areas. Most studies focus on PM10 or PM2.5 (e.g. Soares et al., 2014), which is mainly due to the better established connection to human health effects. Nevertheless, there exist exposure assessments for the cities of Helsinki (Kousa et al., 2002)**
15 **and Oslo (Baklanov et al., 2007), which both focus on the description and application of exposure modeling systems based on chemistry transport modeling. Nevertheless, in our study we used the metric of the annual total population exposure to present our results, while Baklanov et al. (2007) and Kousa et al. (2002) presented exposure averages for different time periods (e.g. afternoons in March) and metrics (e.g. number of persons exposed). Based on the reviewers suggestion, we decided to calculate the averaged afternoon exposure in March 2012, similarly to Kousa et al. (2002), to**
20 **compare the results. Although Kousa et al. (2002) calculated the exposure for March 1996 based on emissions and meteorology specific for Helsinki, we hold this comparison for the meaningful, due to the application of a dynamic population activity with similar time profiles.**

**Thus, we followed Kousa et al. (2002) and calculated the results of the afternoon period (3 p.m.–6 p.m.) in March 2012 for the Riga domain, which is comparable in size and population density. The concentrations, activity and the resulting**
25 **average exposure are calculated as average values in this time period; the corresponding numerical values of activity and average exposure in each grid cell therefore refer to the number of people (density of population), and the concentration times the number of people, respectively. Comparing the results for Riga with the same metric for a similar time shows similar values ranges and maxima for concentrations of NO2, lower population densities per 100 x 100 m² and thus lower exposure values (Figure 1) as presented in Kousa et al. (2002; see figure 3 a-c).**

[Figure]

**Figure 2: The predicted ambient air concentrations of NO2 (ug m-3), the density of population (persons) and the average exposure of the population to NO2 concentrations (ug m-3 * persons), evaluated for the afternoon time period, as an average value in March 2012 in Riga. The grid size is 100m x 100 m, the size of the depicted area is 20km x 20 km.**

5    **Differences in the concentration results can be explained with differences in the input data for city-specific emissions, meteorology and boundary conditions. The lower population density, especially in the centre of the city is mainly an effect of the uniform spatial distribution to Urban Atlas classes, which represent dense city areas but cannot represent real city-specific city centres. This problem will be addressed in future studies by combining OSM points of interest, which can be characterized as city-centre features, to identify areas of high population density during the day. The**

10    **exposure values are the cross product of concentration and population and thus, they are lower than they were calculated in the city of Helsinki. Nevertheless, it appears that exposure values as calculated within our study with the developed generic approach are solid in terms of spatial distribution.**

**Nevertheless, we decided not to integrate this answer into the manuscript, due to the changes in the studies and the resulting comparability, as well as the current extent of the paper.**

*R2.3. "Uncertainties of the NMVOC and the impacts on NOx simulation should also be discussed."*

**Response to R2.3:**

**A section on the uncertainty of NMVOC emissions has been moved from the emission inventory section to the discussion section (see response to R1.1) including a discussion about the impacts of NMOVC on NOx simulation was added. In**

20    **response 1.3 to reviewer 1 we added a revised section of the discussions chapter including uncertainty discussions.**

R2.4. "The uncertainties come from the different steps should be discussed too. For example, the EI, air quality model, exposure responses etc? […], how about the sensitivity of the simulations?"

**Response to R2.4:**

25    **Please see response 1.3 to reviewer 1, in which we added a revised section of the discussions chapter including uncertainty discussions.**

R2.5. "Any seasonal or monthly differences for your results? Or during the shipping busy/non-busy periods […]?"

**Response to R2.5:**

**The scope of this study is the development and application of a generic population exposure modeling approach, which can be applied to different sources of emissions; in this case shipping emissions. Therefore, the detailed description of temporal variation in concentration and exposure results would exceed the scope of this study. Moreover, results of population exposure calculations are mostly used to evaluate long-term health effects (Özkaynak et al., 2013) and therefore this study does not aim at discussing temporal variations. Nevertheless, it would be possible to evaluate seasonal and monthly differences in the concentration and exposure results as well as periods of busy and non-busy shipping activities due to calculated hourly concentrations and exposure. However, this is out of the scope of this study and will be part of future studies.**

**List of Minor changes requested by R2 have been considered in the final manuscript:**

- "Page 1 Line 18, page 12 line 9: 100X100 functions should be used instead of letter x"
  - **We exchanged the letter "x" with "×" for all resolution details in the manuscript.**
- "Page 18: NO2 should use subscript."
  - **We applied subscript for NO2 on page 18.**

[revised manuscript text omitted]